# Molecular variation in a functionally divergent homolog of FCA regulates flowering time in *Arabidopsis thaliana*

Yunhe Wang[1,3], Zhen Tao[1,3], Wanyi Wang[1], Daniele Filiault[2], Chunhong Qiu[1], Chuanhong Wang[1], Hui Wang[1], Shamsur Rehman[1], Jian Shi[1], Yan Zhang[1] & Peijin Li [1]✉

The identification and functional characterization of natural variants in plants are essential for understanding phenotypic adaptation. Here we identify a molecular variation in At2g47310 that contributes to the natural variation in flowering time in *Arabidopsis thaliana* accessions. This gene, which we term *SISTER of FCA* (*SSF*), functions in an antagonistic manner to its close homolog FCA. Genome-wide association analysis screens two major haplotypes of SSF associated with the natural variation in *FLC* expression, and a single polymorphism, SSF-N414D, is identified as a main contributor. The SSF414N protein variant interacts more strongly with CUL1, a component of the E3 ubiquitination complex, than the SSF414D form, mediating differences in SSF protein degradation and *FLC* expression. FCA and SSF appear to have arisen through gene duplication after dicot-monocot divergence, with the SSF-N414D polymorphism emerging relatively recently within *A. thaliana*. This work provides a good example for deciphering the functional importance of natural polymorphisms in different organisms.

[1] The National Engineering Laboratory of Crop Resistance Breeding, School of Life Sciences, Anhui Agricultural University, Hefei 230036, China. [2] Gregor Mendel Institute, Austrian Academy of Sciences, Vienna BioCenter, 1030 Vienna, Austria. [3] These authors contributed equally: Yunhe Wang, Zhen Tao. ✉email: peijin.li@ahau.edu.cn

I n *Arabidopsis thaliana* accessions, there is considerable natural variation in the timing of flowering, a trait important for the evolutionary diversification of this species[1–4]. Differential vernalization requirements were conferred by the floral repressors *FLOWERING LOCUS C* (*FLC*) and *FRIGIDA* (*FRI*), the major factors associated with this natural variation[5–10]. FRI upregulates *FLC* expression, promoting a winter-annual habit, and rapid-cycling behavior has resulted from independent deletions and polymorphisms in *FRIGIDA*[5,7,8,11]. The autonomous promotion pathway-FCA, FPA, FLD, FY, FVE, LD, and FLK-functions antagonistically to *FRI* and represses *FLC* expression[12]. One of the first of these factors to be cloned was FCA, which contains two RNA recognition motifs (RRMs) and, for protein interactions, a WW domain[13]. The WW domain (named for two conserved tryptophans) is required for interaction with other proteins; for example, in budding yeast and mammalian cells, the WW domain in Nedd4 was reported to be required for ubiquitin-proteasome-dependent proteolysis of the bound substrate SPT23[14], and in *A. thaliana*, the WW domain of FCA was reported to mediate the interaction with FY for *FLC* regulation[15].

To investigate the natural variation in flowering time regulation in nature, a large number of genome-wide association studies (GWASs) and quantitative trait locus (QTL) analyses utilizing natural *A. thaliana* accessions have been applied, and numerous candidate genes or genomic regions have been revealed[16–21]. However, the regulatory mechanism of these polymorphisms remains largely elusive, awaiting in-depth study.

The ubiquitin-proteasome system (UPS) is a rapid regulatory mechanism for selective protein degradation and plays a crucial role in the growth and development of different organisms, including flowering time regulation[22,23]. The UPS consists of ubiquitin (Ub), ubiquitin-activating enzyme (E1), ubiquitin-conjugating enzyme (E2), ubiquitin-protein ligase (E3), and the 26S proteasome, which work stepwise and systematically to degrade the target protein substrates[24]. Both genetic and genomic studies have indicated that Ub conjugation is strikingly complex in plants, with more than 1500 Ub-protein ligases (E3s) that direct the final transfer of Ub to specific targets[25,26]. Among these ligases, the CUL-RING ligases (CRLs) are a highly polymorphic E3 collection composed of a CUL backbone that binds carriers of activated Ub and a diverse assortment of adaptors that recruit appropriate substrates for ubiquitylation and degradation[27,28]. In addition, the cullin-associated and neddylation-dissociated 1 (CAND1) protein is involved in one of the most complex regulatory mechanisms affecting CRLs[26,29]. Feng and colleagues have reported that the *cand1* mutant exhibits late flowering and has an increased number of secondary inflorescences[30]; CUL1 has also been shown to be related to the stability of CONSTANS, a crucial factor for flowering time control[31].

In this work, we show a survey of *FLC* expression in a range of selected worldwide accessions (Nordborg set 1 and set2)[32]. This survey identifies molecular variation in At2g47310, a homolog of the autonomous pathway component FCA, as an antagonistic contributor to natural variation in *FLC* and flowering regulation. Our mechanistic analysis demonstrates that the natural polymorphism SSF-N414D affects protein stability and flowering time through distinct interactions with CUL1. The functionally opposing homologs of FCA and SSF appear to have evolved after dicot-monocot divergence, with the At2g47310 polymorphism arising during *A. thaliana* diversification.

## Results

### Identification of *SSF* as a potential contributor to natural variation in flowering time.
For the GWAS analysis, we pre-selected 102 *A. thaliana* accessions that had been shown to express *FLC* at a high level[8,32] and then determined the *FLC* RNA levels before and after cold exposure. We then calculated the vernalization response as the ratio of *FLC* expression in plants exposed to 5 °C for 4 weeks followed by 30 days of growth at 22 °C (T30) to that in nonvernalized plants (NV) (Supplementary Data 1). With these data (T30/NV), we performed a mixed model GWAS with population structure correction. While no SNPs were significantly associated with the phenotype after Bonferroni multiple testing correction (Fig. 1a), we nonetheless looked for potential candidate genes under GWAS peaks that stood out against the background. One peak close to the end of chromosome 2 contained At2g47310, an FCA orthologue that has been proposed to be associated with flowering time[17] (Fig. 1a, Supplementary Table 1). Since population structure correction can potentially overcorrect true associations, we also performed a simple linear model GWAS, where the peak of interest was in fact significant (Supplementary Fig. 1a). While insufficient to definitively demonstrate an association between *FLC* expression and natural variation in At2g47310, we considered this evidence promising enough to pursue At2g47310 as a candidate gene.

We chose to analyze the At2g47310 genomic region (Fig. 1a) and selected accessions carrying the two major haplotypes covering this genomic region: Lov-1 (Lövvik, N. Sweden, latitude 62.5°N) and Ull2-5 (Ullstorp, S. Sweden, latitude 56.06°N). Lov-1 flowered later than Ull2-5 after 4 weeks of vernalization at 5 °C under short-day growth conditions followed by long-day conditions at 22 °C (Supplementary Fig. 1b). In previous research, we performed a QTL analysis of flowering time in the F2 lines generated from a cross between Ull2-5 and Lov-1, which represent the two major haplotypes around At2g47310. We identified three QTL, two on chromosome 5 and one close to the end of chromosome 2[8]. The chromosome 2 QTL explained 10.6% of the flowering time differences, with the Lov-1 allele causing a 19-day increase in flowering time. When these F2 lines were divided by the molecular marker most closely linked to the QTL peak on chromosome 2 for flowering time comparison, the Lov-1 type flowered significantly later than the Ull2-5 type (Fig. 1b). As At2g47310, our candidate from GWAS, was present in the chromosome 2 QTL interval (Supplementary Table 1), we wondered whether *FLC* expression could underlie the flowering time difference between alleles. Indeed, F2 lines carrying the Lov-1 allele of the chromosome 2 QTL had higher *FLC* expression than lines carrying the Ull2-5 allele (Fig. 1c, Supplementary Table 2), consistent with the later flowering of the Lov-1 allele (Fig. 1d). Since all F2 lines included in this analysis had a common genotype at both chromosome 5 QTLs, the observed expression differences were likely primarily attributable to the QTL on chromosome 2.

Sequencing of the coding regions of At2g47310 from the Lov-1 and Ull2-5 accessions uncovered one polymorphism that caused an amino acid change: N to D at position 414. Ull2-5 and Col-0 share the N form, and Lov-1 has the D form (Supplementary Fig. 2a). At2g47310 is annotated as an FCA-like protein (TAIR https://www.arabidopsis.org/) since it is the only homolog in the genome containing both RRM (pfam00076) and WW (pfam00397) domains (Supplementary Fig. 2b, c), so this gene was designated *SSF* (*SISTER of FCA*). Importantly, the single amino acid polymorphism N414D in At2g47310 was located in the WW domain of SSF. The WW domain is considered important for protein interaction[15,33,34]. Therefore, we predicted that the N414D variants in the SSF WW domain might influence the function of SSF by affecting the interaction with other proteins.

### Phenotypic characterization of *ssf* mutants.
To understand the function of *SSF* in *FLC* regulation, we analyzed two independent

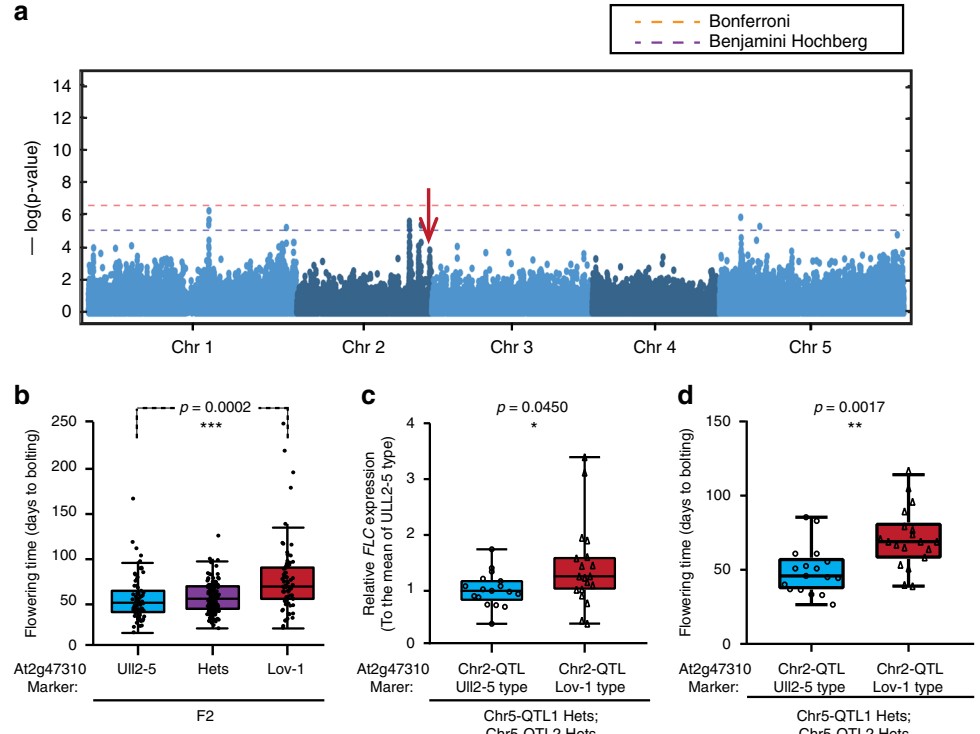

**Fig. 1 Identification of At2g47310 as a potential contributor to flowering time variation. a** Genome-wide association of *FLC* expression (T30/NV) in natural *Arabidopsis* accessions. The vertical arrow at the end of chromosome 2 indicates the position of the gene At2g47310 described in this study. The horizontal dashed lines indicate the significance threshold. **b** Boxplot analysis of the flowering time in F2 individuals from the cross between Ull2-5 and Lov-1[8] (number of F2 individuals: Ull2-5 type = 69, Lov-1 Type = 74 and heterozygous Type = 136). The heterozygous population phenotype is not significantly different from the two parental phenotypes. **c, d** Comparison of *FLC* expression (**c**) and flowering time (**d**) between the F2 lines with Ull2-5 and Lov-1 types of the chromosome 2 QTL under the common Chr5-QTL1 and Chr5-QTL2 background (The QTL positions were shown in our previous research[8], both Chr5-QTL1 and Chr5-QTL2 are heterozygous, number of F2 lines: Chr2-QTL/Ull2-5 Type = 17, Chr2-QTL/Lov-1 Type = 19). The F2 population in (**b, c**) was genotyped and grouped with the molecular marker mostly close to the QTL peaks, respectively. Box plot: lower vertical bar = sample minimum, lower box = lower quartile, middle line = median, upper box = upper quartile, upper vertical bar = sample maximum, single points = outliers. Asterisks indicate significant differences (*$p <$ 0.05, **$p <$ 0.01, ***$p <$ 0.001; two-tailed unpaired *t* test). Source data underlying Fig. 1b–d are provided as a Source data file.

T-DNA insertion lines: Salk_023927C (*ssf-1*) and Salk_028875C (*ssf-2*). These T-DNA insertions, in intron 8 and exon 10 of *SSF*, respectively (Fig. 2a) (Supplementary Fig. 3a, Supplementary Data 2), resulted in the loss of functional alleles, and both mutants showed early flowering and decreased *FLC* expression in both long-day and short-day growth conditions (Fig. 2b–e). Interestingly, we noticed that under long-day growth conditions, the bolting time and leaf number differences between *ssf* and Col-0 were comparably match-able, whereas under short-day growth conditions, *ssf* mutants flowered ~25 days earlier than Col-0, but the leaf number difference was only 4 leaves, which implied that the leaf initiation rate might be higher in *ssf* than wild type under short-day conditions and a putative complex nature of the molecular mechanism underlying the *ssf* phenotype (Fig. 2b, c, Supplementary Fig. 3b–e). The early flowering was surprising given that *SSF* is a homolog of FCA, and *fca* mutants exhibit very late flowering[35]. *FLC* silencing was also attenuated in *ssf* following 4 weeks of vernalization, accounting for the GWAS signal (Fig. 2f, g). We chose to focus on the early-flowering phenotype of *ssf* under nonvernalizing conditions.

**SSF directly binds *FLC* chromatin and regulates RNA Pol II accumulation in *FLC* genomic regions.** To investigate the relationship between SSF and other pathways involved in *FLC* regulation and flowering time control, we carried out genetic analyses. *ssf-2* was crossed with a FRIGIDA+ line and the defective autonomous pathway mutant *fca-9*. *FLC* expression was analyzed and compared with the corresponding single mutants. The double mutants FRIGIDA *ssf-2* and *fca-9 ssf-2* exhibited intermediate phenotypes, slightly lower *FLC* expression than the late-flowering single mutants, and much higher *FLC* expression than *ssf-2* alone (Fig. 2h). The flowering time of these genotypes was also determined. FRIGIDA *ssf-2* flowered earlier than the corresponding controls, but *fca-9 ssf-2* flowered similarly to *fca-9* (Supplementary Fig. 3f, Supplementary Data 2). The inconsistency between *FLC* expression and flowering time between *fca-9* and *fca-9 ssf-2* suggested that other flowering regulators were also affected by the *ssf* mutation.

To further reveal how SSF affects *FLC* expression, we analyzed unspliced *FLC* expression and found that consistent with the change in mature *FLC*, the expression level of unspliced *FLC* was also lower in the *ssf* mutant than in wild-type Col-0 (Supplementary Fig. 3g). These results suggested that SSF might influence *FLC* expression at the gene transcriptional level. A gene transcription test was then performed to validate this prediction, and a construct that fused the 1.6-kb-long *FLC* promoter with the firefly luciferase (LUC) coding sequence was prepared and cotransformed together with a SSF overexpression (SSF-OE) construct into *Nicotiana benthamiana* leaves. The LUC signal was compared with that in leaves transformed without SSF-OE. The enzyme activity test and fluorescence comparison showed that SSF significantly promoted *FLC* promoter-driven *LUC* expression (Fig. 2i, Supplementary Fig. 3h), and consistent with this finding, RNA polymerase II (Pol II) was significantly less enriched in the

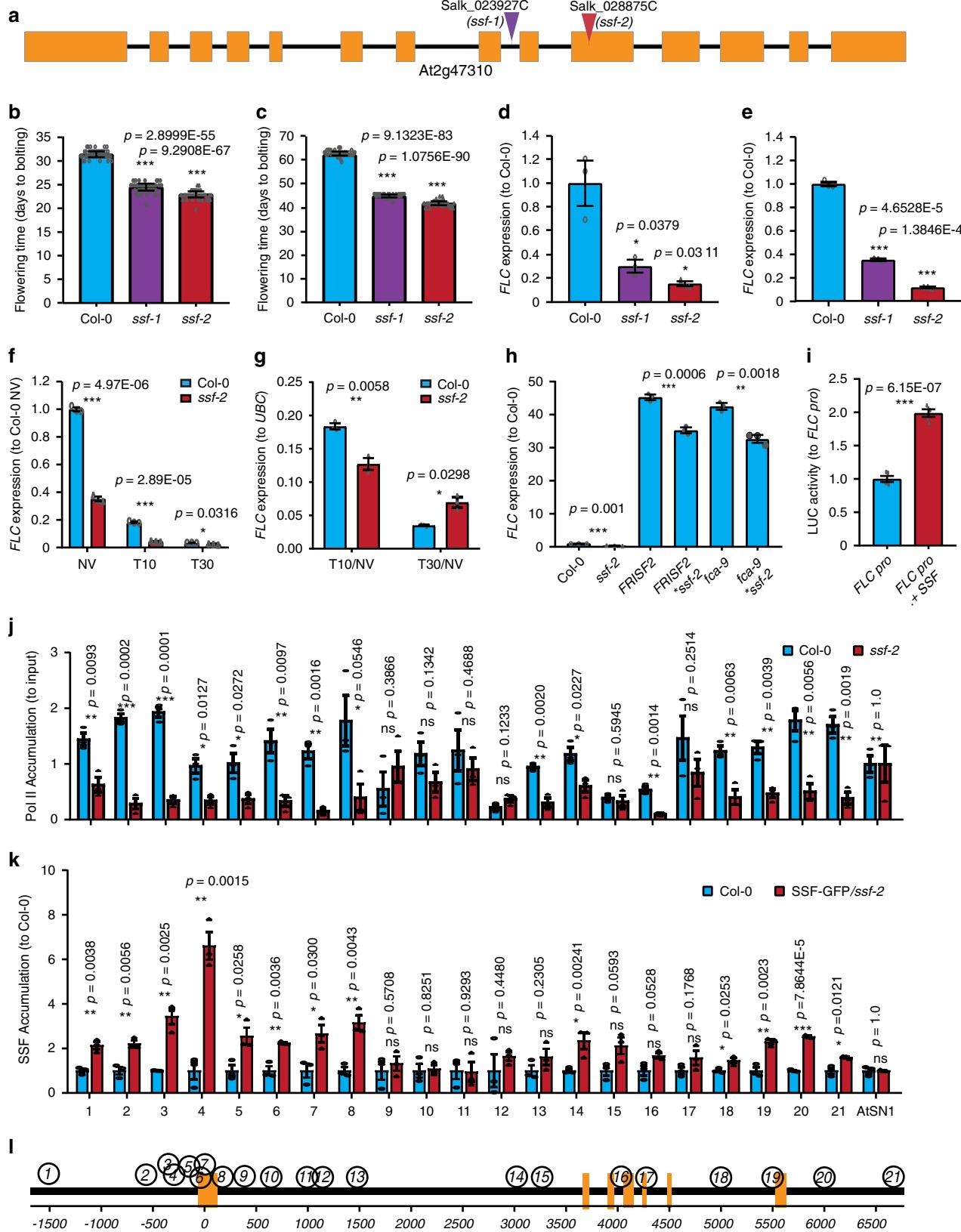

*ssf-2* mutant at the *FLC* promoter, gene body, and 3′ untranslated regions (UTRs) (Fig. 2j, l). Furthermore, in vivo chromatin immunoprecipitation (ChIP) assays using SSF-GFP transgenic plants, which fully rescued the *ssf-2* phenotype (Supplementary Fig. 4a, b), showed that SSF was significantly enriched at the promoter, gene body, and 3′ UTRs of the *FLC* genome, indicating that SSF could bind *FLC* chromatin and directly regulate *FLC* expression (Fig. 2k, l). Taken together, these results suggest that SSF functions as a positive regulator of *FLC* transcription, in contrast to its homolog FCA.

**Fig. 2 Characterization of At2g47310 *ssf* mutants. a** Gene structure of At2g47310 and the T-DNA insertion sites in the two independent *ssf* mutants. Yellow boxes represent exons, and horizontal lines represent introns. **b**, **c** Comparison of flowering time between the *ssf* mutants and wild-type Col-0 under long-day (**b**) and short-day (**c**) growth conditions ($n = 48$). **d**, **e** *FLC* expression is downregulated in *ssf* mutants compared with Col-0 under long-day (**d**) and short-day (**e**) growth conditions. Total RNA was extracted from seedlings after two weeks of germination under long-day or short-day growth conditions ($n = 3$). **f** *FLC* expression in response to 4 weeks of vernalization ($n = 3$). **g** Relative *FLC* expression before and after vernalization ($n = 3$). In (**f**, **g**), for NV, seedlings were collected after two weeks of growth under long-day growth conditions; T10 and T30 samples were grown under long-day conditions, vernalized for 4 weeks, and returned to normal conditions for 10 and 30 days for sample collection and RNA extraction. **h** *FLC* expression of double mutants under long-day growth conditions ($n = 3$). **i** Comparison of LUC activity between the *FLCpro::LUC* transformants with and without *SSF-OE* ($n = 5$). *FLCpro::LUC* with *SSF-OE* or with empty vector was cotransformed into tobacco leaves. R-LUC was used as an internal control. **j** Accumulation of Pol II normalized to input in different regions of *FLC* ($n = 3$). **k** Occupancy of SSF-GFP normalized to input and Col-0 in different regions of *FLC* ($n = 3$). **l** Schematic presentation showing the position of the qPCR primers in (**j**, **k**) in *FLC* genomic regions. In (**b**, **k**), data are presented as mean ± SEM. Asterisks indicate significant differences (*$p < 0.05$; **$p < 0.01$; ***$p < 0.001$; ns, not significant; two-tailed unpaired $t$ test). Source data underlying Fig. 2b–k are provided as a Source data file.

**Identification and functional validation of SSF-N414D as a causative natural polymorphism for *FLC* expression.** Five SNPs are present in the *SSF* alleles between the parental lines used in the QTL analysis (Ull2-5 and Lov-1): three SNPs located in introns, one SNP in exon 2 that does not change the amino acid, and one nonsynonymous natural polymorphism N414D in exon 11, with Col-0 sharing the same 414N polymorphism as Ull2-5 (Supplementary Fig. 2a). Site-directed mutagenesis was applied to convert the Col-*SSF* allele with 414N to the D form, and the mutant and wild-type constructs driven by the native *SSF* promoter plus the 3′ UTR were transferred to a binary vector for gene transformation in the *ssf-2* mutant (Fig. 3a). T2 transgenic plants were grown in soil under long-day growth conditions at 22 °C, and their flowering time was analyzed. Due to considerable variation between independent transgene lines, as found in our previous study[11], we assayed a large set of transgenic plants and compared the flowering times of 23 lines with Col-SSF414N and 54 lines with the Col-SSF414D construct. Overall, the Col-SSF414D transgenic plants flowered significantly later than those with Col-SSF414N (Fig. 3b). Then, additional transgenic plants (in total, 72 lines for Col-SSF414N and 63 lines for Col-SSF414D) were generated for *FLC* expression analysis. Equal amounts of RNA from 50 independent T2 transgenic lines[11,36] were pooled and subjected to a quantitative analysis of *FLC* expression. The 414N to 414D change increased *FLC* expression (Fig. 3c, Supplementary Table 3), resulting in later flowering and explaining why this polymorphism was found in the GWAS mapping variation (Figs. 1a, 3b). Two representative transgenic lines with similar *SSF* expression levels from Col-SSF414N and -414D were further selected for the *FLC* expression and flowering time assays, and the results indicated that SSF414D caused higher *FLC* expression and later flowering than SSF414N (Fig. 3d–f). Therefore, we conclude that the SSF-N414D molecular change is an important contributor to the QTL at the end of chromosome 2 responsible for the phenotypic variation in *FLC* expression and flowering time.

**CUL1 interacts with SSF and contributes to flowering time regulation by promoting SSF degradation.** To further dissect the mechanism underlying SSF function, we performed a series of protein pulldown assays plus mass spectrometry analysis using the two types of transgenic plants, Col-SSF414N-GFP and Col-SSF414D-GFP, which fully rescued the *ssf-2* flowering time phenotype (Supplementary Fig. 4a, b). Multiple UPS-related factors were detected in the SSF protein complex, including CUL1 (At4g02570) and CAND1 (At2g02560), the two major components of the E3 ubiquitination ligase of the UPS (Supplementary Table 4). In the two types of samples, peptide enrichment of GFP relative to SSF from SSF414N-GFP and SSF414D-GFP was similar, but a higher amount of CUL1 and CAND1 peptides was

pulled down by SSF414N than by SSF414D (Supplementary Fig. 4c–e). CUL1 was selected for further functional analysis, and 4 approaches were utilized to verify the interaction between SSF and CUL1. First, a yeast two-hybrid (Y2H) assay was carried out by fusing the SSF coding region in-frame with the binding domain (BD) and CUL1 with the activation domain (AD) in the Y2H vectors. The results indicated that both SSF414N and SSF414D interacted with CUL1, and although initially the same amount of yeast grew similarly on SD medium lacking leucine and tryptophan (Supplementary Fig. 4f), SSF414N interacted with CUL1 more strongly than SSF414D in SD medium lacking leucine, tryptophan, histidine, and adenine (Fig. 4a). Second, the bimolecular fluorescence complementation (BiFC) technique was applied and showed that both SSF414N and 414D could interact with CUL1 in the nuclei of *A. thaliana* protoplasts (Fig. 4b). Third, the SSF protein with a His tag was expressed in *Escherichia coli* cells, purified, and incubated with CUL1 with a Flag tag expressed in *N. benthamiana* for in vitro pulldown analysis, which showed that both SSF414N and 414D could form a protein complex with CUL1 (Fig. 4c, Supplementary Fig. 4g, h). Finally, GFP-tagged SSF variants and Flag-tagged CUL1 were transformed into *A. thaliana* protoplasts for a coimmunoprecipitation (Co-IP) assay, and the results confirmed the in vivo interaction between these two proteins (Fig. 4d).

The interaction between SSF and CUL1 prompted us to check whether SSF could be a substrate of the UPS and was regulated by the UPS at the protein level. Consistent with our expectation, when SSF-GFP was cotransformed into protoplasts together with an empty vector or a CUL1 overexpression (CUL1-OE) vector and the GFP signal was normalized to the fluorescence signal of chlorophyll from each cell, the SSF-GFP signal with CUL1 was significantly weaker than those with SSF-GFP and the empty vector (Fig. 5a, b). We also wanted to further check the SSF level in the *cul1* mutant. As a *cul1* null mutant is embryonically lethal[37], a weaker mutation containing a T-DNA insertion in the promoter region of the CUL1 gene was obtained from NASC. In the homozygous mutant *cul1-1* (SALK_204945C), *CUL1* expression was slightly but significantly knocked down compared with wild-type Col-0 (Supplementary Fig. 5a–c). Consistent with the CUL1-OE experiments, the signal of SSF-GFP was significantly stronger in the transformant of the *cul1-1* mutant than in wild-type Col-0 (Fig. 5c, d).

To further confirm the role of CUL1 in the functioning of SSF, we checked the His-tagged SSF protein incubated in protein extracts from wild-type Col-0 and the *cul1-1* mutant. The results of this cell-free degradation assay showed that the degradation rate of SSF in the *cul1-1* mutant was significantly slower than in the wild type (Fig. 5e, f). When the proteasome inhibitor MG132 (Z-Leu-Leu-Leu-CHO) was included in the assays, SSF degradation was eliminated (Fig. 5e, f). These results indicated that CUL1

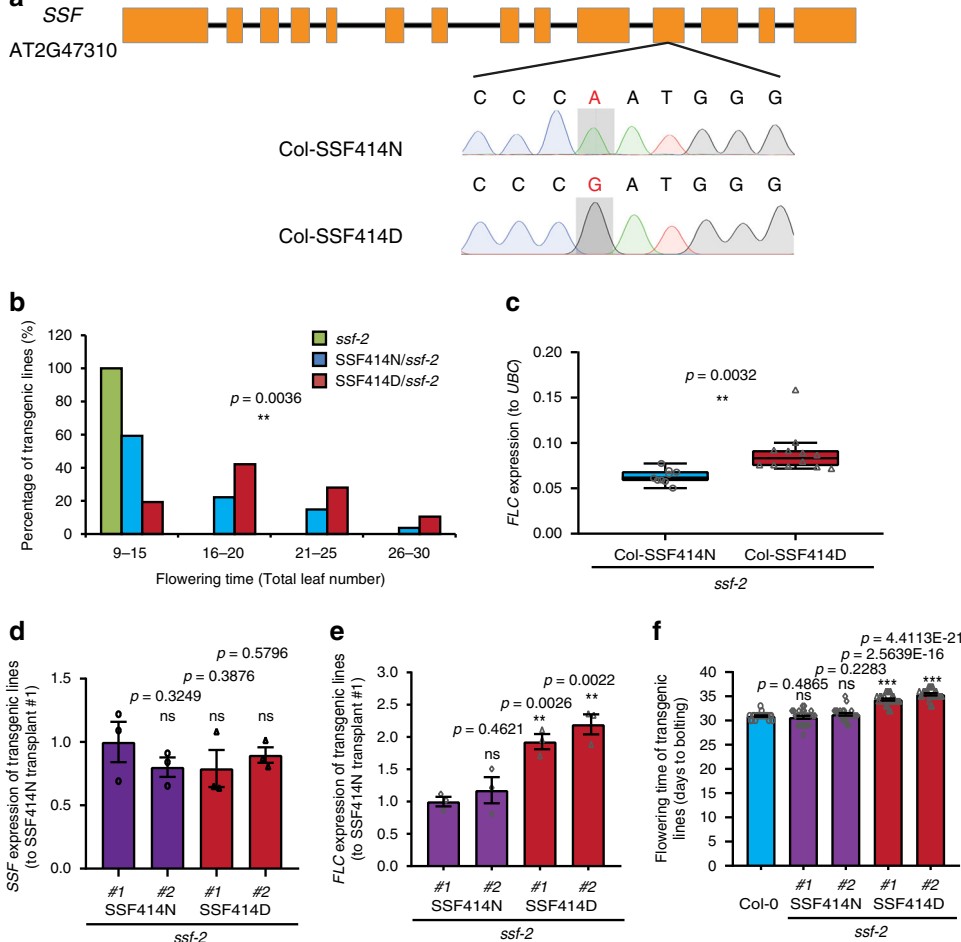

**Fig. 3 The SSF-N414D polymorphism affects flowering time and *FLC* expression. a** Schematic illustration of *SSF* gene structure and the sequencing results of N414D mutagenesis. Boxes represent exons, and horizontal lines indicate introns. The polymorphisms of A/G nucleotides are shown in red. **b** Flowering time comparison of Col-SSF414N/*ssf-2* and Col-SSF414D/*ssf-2* transgenic plants. Twenty-three and 54 independent T2 transgenic lines were scored, with eight individuals from each line included. The number for *ssf-2* was 20. **c** Comparison of *FLC* expression from the two transgenic plant materials containing either Col-SF414N/*ssf-2* (number of pools = 9) or Col-SSF414D/*ssf-2* (number of pools = 12). Fifty independent T2 transgenic lines from each construct were mixed to generate each sample pool. The difference in *FLC* expression between the two types of transgenic plants was significant. The *UBC* gene was used as an internal control for the RT-qPCR assays. Box plot: lower vertical bar = sample minimum, lower box = lower quartile, middle line = median, upper box = upper quartile, upper vertical bar = sample maximum, single points = outliers. **d–f** Two representative independent transgenic lines of Col-SSF414N and 414D with similar transgene expression (**d**) show distinct *FLC* expression levels (**e**) and flowering times (**f**) (n = 24). Data are presented as mean ± SEM. Asterisks indicate significant differences (**p < 0.01; ***p < 0.001; ns, not significant; two-tailed unpaired *t* test). Source data underlying Fig. 3b–f are provided as a Source data file.

could promote SSF protein degradation. Under long-day growth conditions, the *cul1-1* mutant flowered earlier than Col-0 (Fig. 5g), and consistent with this result, *FLC* expression was downregulated and *FT* was upregulated in the *cul1-1* mutant (Supplementary Fig. 5d, e). The flowering time of the *cul1-1* mutant and the expression of *FLC* and *FT* in this mutant were not consistent with our expectations, since CUL1 could degrade the flowering time repressor SSF, and in theory, *cul1-1* mutation should lead to late flowering. We reasoned that since CUL1 is a general E3 ubiquitin ligase in plants, it may affect various aspects of plant development; for example, it has been reported to be related to auxin regulation[38], so it is possible that in addition to regulating SSF protein degradation, CUL1 may regulate *FLC* expression and flowering time through other components such as FCA, a flowering inducer containing a WW domain similar to SSF[13,15]. Consistent with this prediction, the Y2H assay, fusing the FCA coding region in-frame with the BD and CUL1 with the AD in the Y2H vectors, indicated that FCA interacted with CUL1

in yeast (Supplementary Fig. 6). We then used an FCA antibody and performed a western blot assay for wild-type Col-0 and the *cul1-1* mutant with and without MG132 treatment. The results showed that the FCA protein level was increased significantly in the *cul1-1* mutant compared with Col-0, and MG132 treatment caused higher FCA accumulation than the control (Fig. 5h, i), suggesting that FCA was also a target of CUL1 for protein degradation.

In addition, as expected, the expression of SSF and FCA was not influenced by the *cul1-1* mutation, reinforcing the notion that the regulatory effect of CUL1 on SSF and FCA occurred at the protein level (Supplementary Fig. 5f, g).

**The N414D polymorphism affects the stability of the SSF protein**. To check the involvement of the SSF-N414D polymorphism in UPS-related protein degradation of SSF, we expressed and purified SSF414N and SSF414D proteins with His tags in the *E. coli* system and incubated the protein with Col-0

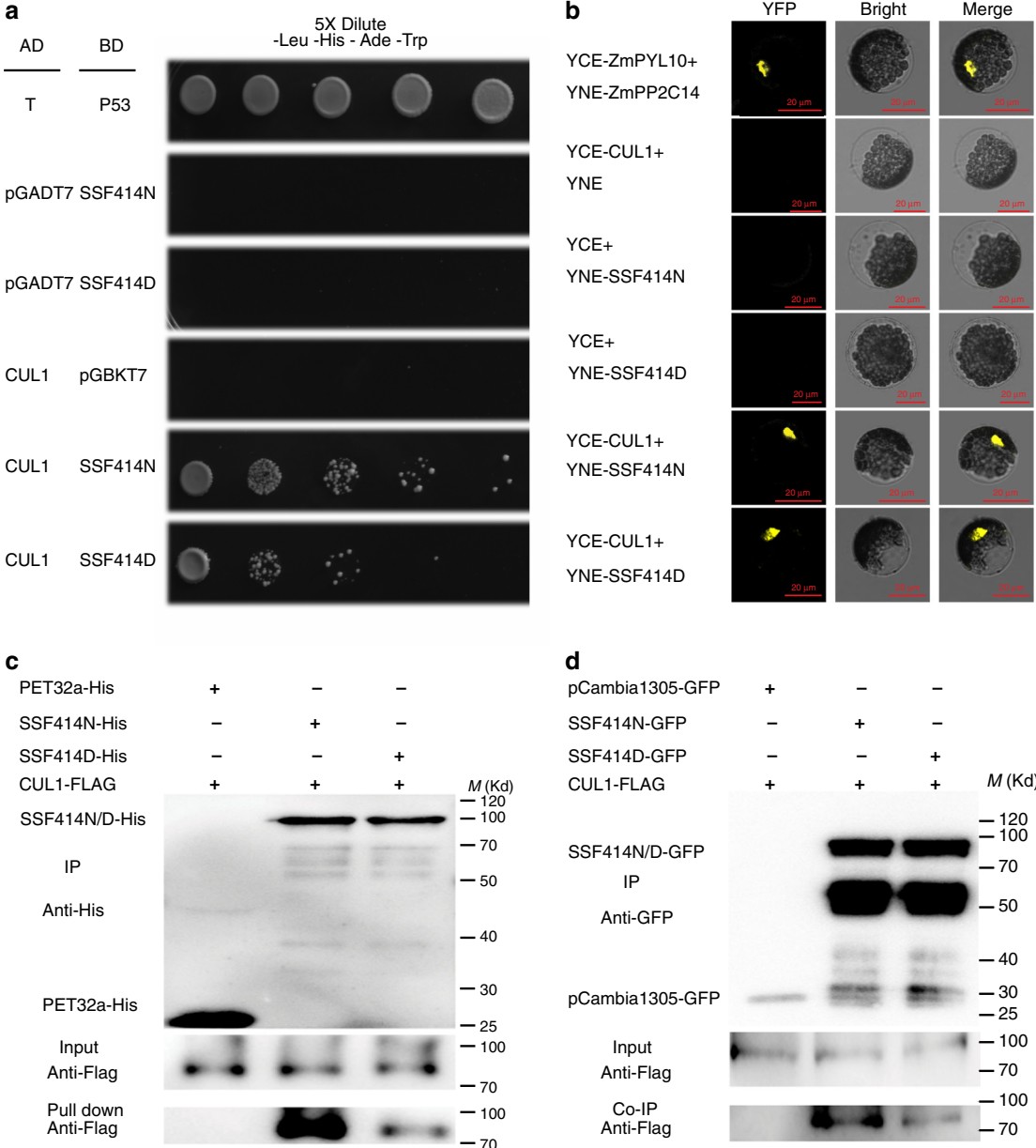

**Fig. 4 CUL1 interacts with SSF in vitro and in vivo. a** Yeast two-hybrid analysis of the interaction between SSF414N, 414D, and CUL1. The yeast culture was quantified and diluted in a 5× series and grown on YEP medium without leucine, tryptophan, adenine, and histidine. The same amount of yeast cells was grown from the start in each assay. The interaction of T and P53 was used as a positive control. The interactions between the empty vector AD with SSF414N or SSF414D and the empty vector BK with AD-CUL1 were used as negative controls. **b** SSF414N or SSF414D fused to the N-terminus of YFP was tested for its ability to bind to CUL1 fused to the C-terminus of YFP. Yellow fluorescence and a bright-field image were recorded and merged. The N-terminus of YFP alone plus CUL1 fused to the C-terminus of YFP and the C-terminus of YFP plus SSF414D or 414N fused to the N-terminus of YFP were used as negative controls. The interaction of ZmPYL10 and ZmPP2C14 was used as a positive control. Twenty cells were examined for each transformation. **c** Beads containing a His tag or His-fused SSF414N or SSF414D were tested for binding of Flag-fused CUL1 expressed in tobacco. The bound proteins were immunoblotted with anti-Flag antibody. pET32a is shown as a negative control. **d** Co-IP analysis of SSF414N-GFP or SSF414D-GFP cotransformed with Flag-CUL1 into *Arabidopsis* protoplasts. The expressed proteins were immunoprecipitated using GFP-Trap beads and then detected with anti-Flag antibody. pCambia1305-GFP is shown as a negative control. These experiments were repeated at least three times, and the same results were obtained. Source data are provided as a Source data file.

tissue extract. This protein degradation assay indicated that Col-SSF414N degraded quickly with incubation, but in contrast, Col-SSF414D degraded at a significantly lower rate. When the proteasome inhibitor MG132 was included in the incubation, the difference in SSF degradation was eliminated (Fig. 6a, b).

To confirm the role of SSF-N414D on SSF stability in vivo, we checked *ssf* transgenic plants expressing Col-SSF414N-GFP and Col-SSF414D-GFP under the control of the native SSF promoter and compared the GFP signals in multiple SSF414N or D-GFP transgenic lines, which fully rescued the *ssf-2* flowering time phenotype (Supplementary Fig. 4a, b). Comprehensive confocal imaging showed that SSF414D-GFP accumulated to a greater extent than SSF414N-GFP relative to the fluorescence of DAPI (4′,6-diamidino-2-phenylindole)-stained nuclei, and when MG132 was

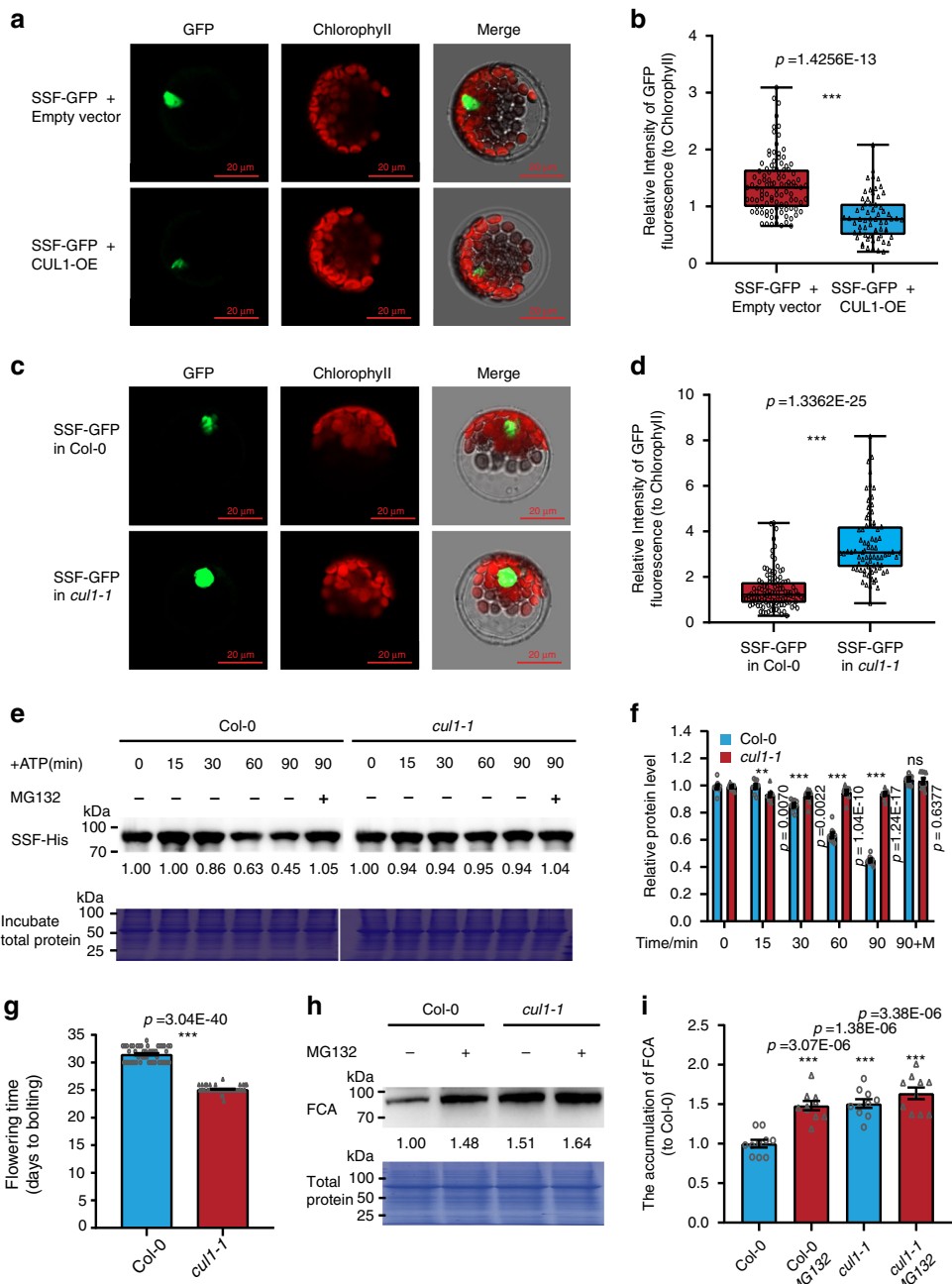

**Fig. 5 CUL1 could promote SSF degradation. a, b** The SSF-GFP signal was downregulated by overexpression of *CUL1* (*CUL1-OE*) in *Arabidopsis* protoplasts. The GFP signal was quantified and normalized to chlorophyll fluorescence in 102 cells for SSF-GFP + Empty vector and 63 cells for SSF-GFP + CUL1-OE, and each type was randomly collected under confocal microscopy (**b**). Observation and quantification of each cell were performed under the same set of confocal parameters and at the same scale. **c, d** The SSF-GFP signal was upregulated in *cul1-1* protoplasts compared with Col-0 protoplasts. The SSF-GFP signal from 77 cells of each type was quantified and normalized to chlorophyll fluorescence (**d**). The results in (**a–d**) were confirmed at least three times. Bar: 20 μm. **e, f** The degradation of SSF was slower in *cul1-1* than Col-0 in the cell-free protein degradation assay. Recombinant purified His-SSF was incubated in equal amounts of total proteins extracted from 14-day-old wild-type Col-0 and *cul1-1* seedlings in the presence of ATP. Representative images are shown in (**e**), and relative protein levels are shown in (**f**). His-SSF was detected with an anti-His antibody. Total protein was used as a loading control. In (**f**), $n = 10$ independent replicates. **g** Comparison of flowering time between *cul1-1* and wild-type Col-0 under long-day growth conditions ($n = 48$). **h, i** FCA is more enriched in *cul1-1* than Col-0. Total protein was extracted from 14-day-old Col-0 and *cul1-1* seedlings. FCA was detected with an anti-FCA antibody. Total protein was used as a control. Representative images are shown in (**h**), and relative protein levels are shown in (**i**). In (**i**), $n = 10$ independent replicates. These experiments were repeated at least three times, and consistent results were obtained. In (**b, d**), Box plot: lower vertical bar = sample minimum, lower box = lower quartile, middle line = median, upper box = upper quartile, upper vertical bar = sample maximum, single points = outliers. In (**f, g, i**), data are presented as mean ± SEM, and asterisks indicate significant differences (*$p < 0.05$; **$p < 0.01$; ***$p < 0.001$, ns, not significant; two-tailed unpaired *t* test). Source data are provided as a Source data file.

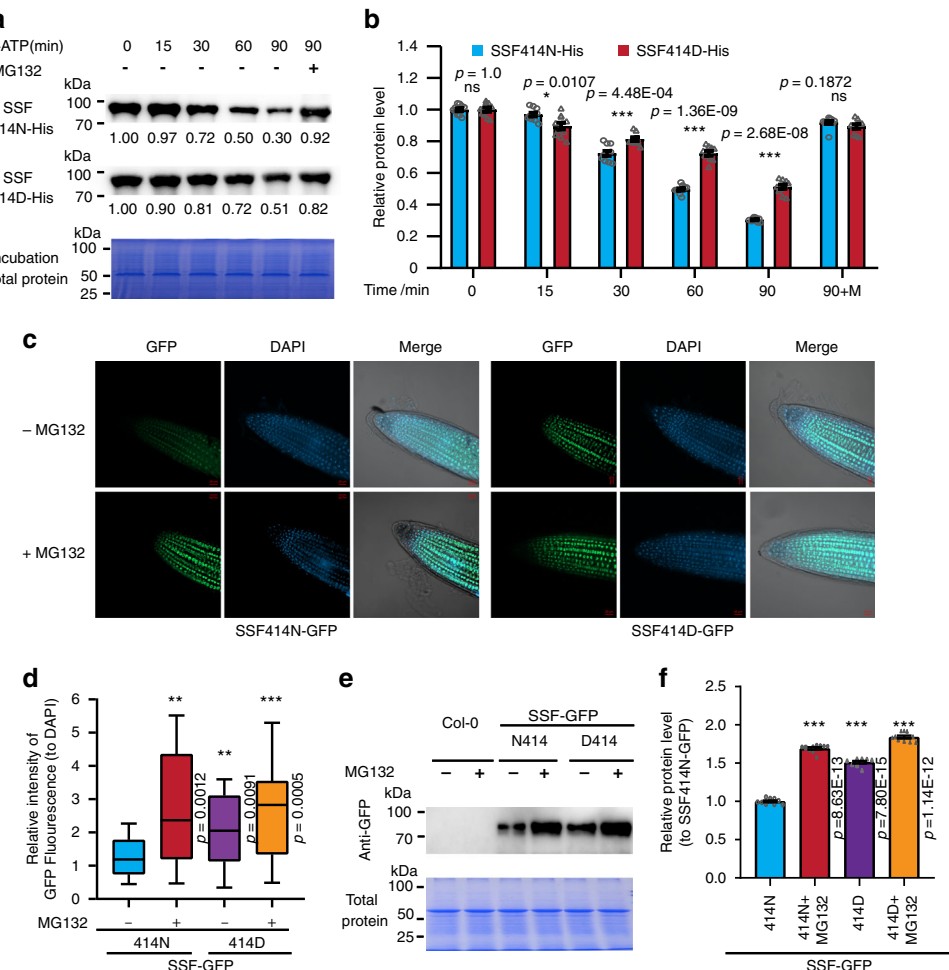

**Fig. 6 The natural polymorphism SSF-N414D affects protein stability through the ubiquitination pathway. a**, **b** The degradation of SSF414D was slower than that of SSF414N in the cell-free protein degradation assay. Recombinant purified His-SSF414N/414D was incubated in equal amounts of total proteins extracted from 14-day-old wild-type Col-0 seedlings in the presence of ATP. His-SSF414N/414D was detected with an anti-His antibody. Representative images are shown in (**a**), and relative protein levels are shown in (**b**). In (**b**), data are presented as mean ± SEM, and $n = 10$ independent replicates. **c**, **d** Stability of SSF414N/414D-GFP in vivo. Fourteen-day-old SSF414N-GFP and SSF414D-GFP transgenic seedlings were treated with 50 μM MG132. In (**c**, **d**), SSF-GFP- and DAPI-stained nuclear signals in roots of 18 independent lines were collected by confocal microscopy and quantitatively analyzed under the same set of fluorescence intensities and at the same scale. From each line, more than 10 seedlings were assayed. Representative pictures are shown in (**c**), and the relative GFP signal level is normalized to DAPI in (**d**). In (**d**), box plot: lower vertical bar = sample minimum, lower box = lower quartile, middle line = median, upper box = upper quartile, upper vertical bar = sample maximum, single points = outliers. **e**, **f** Detection of SSF-GFP in vivo from Col-0, SSF414N-GFP, and SSF414D-GFP plants. Total protein was extracted from whole seedlings of 14-day-old Col-0, SSF414N-GFP, and SSF414D-GFP plants, incubated with GFP-trap to enrich GFP-fused protein, and analyzed by immunoblotting using an anti-GFP antibody. Total protein was used as the input loading control. In (**f**), data are presented as mean ± SEM, and $n = 10$ independent replicates. These experiments were repeated at least three times, and consistent results were obtained. In (**b**, **d**, **f**), asterisks indicate significant differences (*$p < 0.05$; **$p < 0.01$; ***$p < 0.001$; ns, not significant; two-tailed unpaired $t$ test). Source data are provided as a Source data file.

included in the assays, the SSF414N level was increased to a level equivalent to that of SSF414D-GFP (Fig. 6c, d). This finding was further verified by western blot analysis using a GFP antibody (Fig. 6e, f). Moreover, the ubiquitination and SSF levels of the SSF414N-GFP and SSF414D-GFP proteins in transgenic plants were compared using an Ubi antibody, and SSF414N-GFP and 414D-GFP, which have similar transgene *SSF* expression levels, were selected for SSF protein and flowering time comparison (Supplementary Fig. 7a). The results indicated that the ubiquitinated SSF was more abundant in SSF414N-GFP than SSF414D-GFP plants, and consistent with this finding, SSF414N was degraded to a greater extent than SSF414D (Supplementary Fig. 7b). Taken together, this evidence indicated that the N414D polymorphism affected *FLC* expression and flowering time by regulating SSF protein stability in a UPS/CUL1-dependent manner.

**Geographic distribution of SSF-N414D groups and phylogenetic analysis of SSF.** We extracted the available *A. thaliana* SSF protein sequences from a larger accession sequence set (1001 Genomes database) and compared their sequences to the geographical collection sites of the accessions (http://1001genomes.org). The 414N and D allele types fell into two main SSF groups (Supplementary Fig. 8), which showed a distinct geographical pattern in Sweden (Fig. 7a); 414N occurred predominantly in S. Sweden, while 414D occurred mainly in N. Sweden, and no clear latitudinal pattern was observed in Spain or Germany (Fig. 7b, c).

We also wanted to explore the functional differentiation between SSF and FCA to investigate the evolutionary occurrence of SSF and FCA in different plant species. In *A. thaliana*, SSF and FCA were the only two proteins having both RRM and WW domains, suggesting a close evolutionary relationship. The genetic

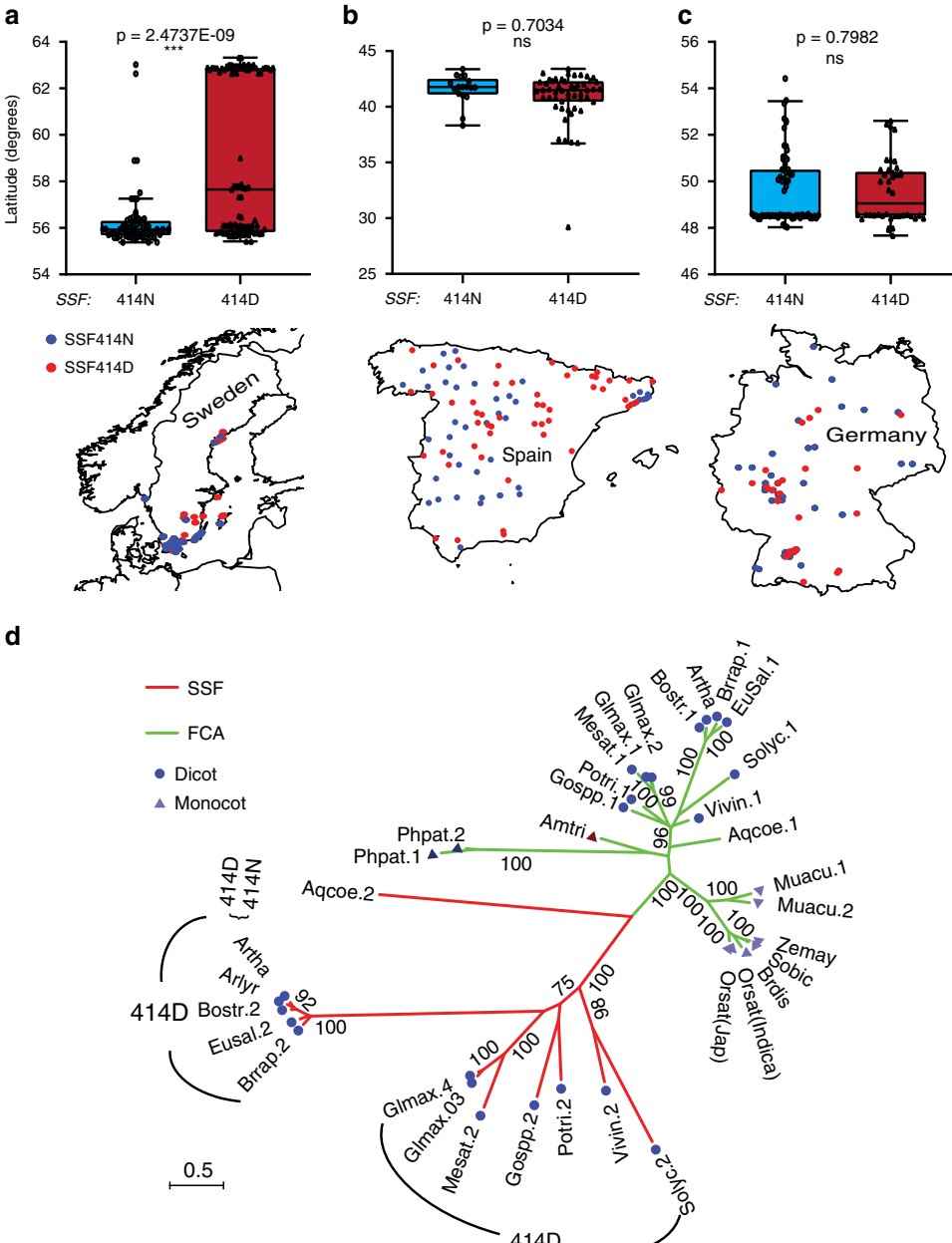

**Fig. 7 Geographical distribution of the SSF-N414D polymorphism in natural *Arabidopsis* accessions and phylogenetic analysis of SSF homologs. a–c** Geographical distribution of the SSF-N414D polymorphism. The accessions from Sweden (**a**, n = 167), Spain (**b**, n = 98), and Germany (**c**, n = 108) were divided into two groups based on the polymorphism SSF-N414D and according to the latitude of the collection site. The maps and accession distribution are shown underneath. Box plot: lower vertical bar = sample minimum, lower box = lower quartile, middle line = median, upper box = upper quartile, upper vertical bar = sample maximum, single points = outliers. **d** Phylogenetic analysis of SSF homologs in different plant species. The red and green lines indicate SSF- and FCA-type proteins, respectively, the purple triangles indicate monocot plants, and blue circles indicate dicot plants. The numbers along with tree branches are the bootstrap values with 100 replications. Amtri: *Amborella trichopoda*; Aqcoe: *Aquilegia coerelea*; Arlyr: *Arabidopsis lyrata*; Artha: *Arabidopsis thaliana*; Bostr: *Boechera stricta*; Brdis: *Brachypodium distachyon*; Brrap: *Brassica rapa*; Eusal: *Eutremasal sugineum*; Glmax: *Glycine max*; Gospp: *Gossypium spp*; Mesat: *Medicago sativa*; Muacu: *Musa acuminata*; Orsat: *Oryza sativa*; Phpat: *Physcomitrella patens*; Potri: *Populus trichocarpa*; Sobic: *Sorghum bicolor*; Solyc: *Solanum lycopersicum*; Vivin: *Vitis vinifera*; Zemay: *Zea mays*. Asterisks indicate significant differences (***p < 0.001, ns, not significant; two-tailed unpaired *t* test). Source data underlying Fig. 7a–c are provided as a Source data file.

divergence of SSF proteins was significantly higher than that of FCA (Fig. 7d, Supplementary Fig. 9, Supplementary Data 3, 4), and importantly, SSF-type genes were found only in dicot genomes. In addition, *Arabidopsis lyrata* and *Brassica oleracea* had the 414D form, indicating that the 414N polymorphism was the more recently evolved form. In contrast, FCA-type genes were present in both dicot and monocot genomes, including in

*Amborella trichopoda* (Amtri), *Physcomitrella patens* (Phpat 1, 2), and *Aquilegia coerelea* (Aqcoe 1, 2).

## Discussion
Considerable effort has been devoted to the identification of naturally occurring polymorphisms underpinning phenotypic

change, as they are the key to deciphering the molecular basis of evolution and adaptation[39,40]. In *A. thaliana*, an important adaptive trait is flowering time[41], and this trait has been analyzed in many GWASs and QTL studies[17,19]. Here we found that the variation in *SSF* regulated *FLC* expression and flowering time through a mechanism connected to UPS-mediated protein degradation.

Based on the annotation of the protein sequence, *SSF* was found to encode a homolog of FCA, and in the TAIR database, it is annotated as an FCA-like protein. Interestingly, SSF had an antagonistic effect on flowering and *FLC* expression compared with FCA; under both long-day and short-day photoperiod growth conditions, *ssf* mutants showed early flowering (Fig. 2a–c), whereas *fca* mutants showed late flowering[35]. Further genetic analysis with double mutants indicated that SSF functioned independently of FCA at the *FLC* regulation level (Fig. 2h). SSF contained two RRM domains, which helped us dissect the mechanism of SSF function. Our data indicate that SSF binds the promoter region, gene body, and the 3′ UTRs of the *FLC* genome (Fig. 2k) and promotes *FLC* transcription (Fig. 2i) by recruiting the RNA Pol II machinery (Fig. 2j). The SSF binding sites in *FLC* differed from those in FCA (Fig. 2k)[35], which suggested that these two homologous factors might have different mechanisms of *FLC* regulation.

Natural variation has become a research focus in recent years, being used to reveal the evolutionary history and functional importance of genes in various organisms[40,42], and many natural polymorphisms associated with phenotypes have been detected[17,39,43]; however, in most cases, the mechanism responsible for these polymorphisms remains poorly understood. In this study, we identified the natural polymorphism SSF-N414D and demonstrated that SSF and the N414D polymorphism function through interacting with CUL1, one of the major components of the UPS. We provided several lines of evidence to illustrate the interaction and functional importance. (1) From mass spectroscopy experiments with the transgenic plant containing GFP-tagged SSF variants (Supplementary Table 4), we identified several components of the UPS, including CUL1, characterized in this study, and this interaction was further validated with in vitro and in vivo Y2H, BiFC, and Co-IP experiments (Fig. 4a–d). (2) Treatment with the proteasome inhibitor MG132 and in vitro protein degradation assays indicated that SSF was ubiquitinated and degraded in a manner dependent on CUL1 (Fig. 5e, f). (3) More importantly, CUL1 could recognize and distinguish the SSF-N414D polymorphism in the WW domain, which led to differential protein degradation between SSF414N and 414D variants (Fig. 6). Given that CUL1 degrades SSF, which enhances *FLC* expression and represses flowering, we predicted that the *cul1-1* mutant would show late flowering, but in contrast to this prediction, the *cul1-1* mutant showed earlier bolting, decreased *FLC* expression, and increased *FT* expression (Fig. 5g, Supplementary Fig. 5d, e). We reasoned that in addition to the regulation of SSF, CUL1 might influence flowering time in different ways, for example, by targeting different targets. Consistent with this explanation, CUL1 has been reported to interact with CONSTANS, an important factor for flowering time control[31], and we found in this study that CUL1 could interact with and degrade the SSF homolog FCA (Fig. 5h, i, Supplementary Fig. 6), a strong flowering inducer, which explains the early-flowering phenotype of the *cul1-1* mutant.

Comparison with *A. lyrata* SSF, which carries the D form, suggested that the N form arose relatively recently in *A. thaliana* (Fig. 7d). In Sweden, accessions with SSF414N exist predominantly in the south, and those with SSF414D are present mainly in the north (Fig. 7a). One particular *FLC* haplotype (Slow vernalizer, SV1) shows a similarly nonrandom latitudinal distribution[11]. This haplotype contains an intronic SNP that influences splicing of the distal *COOLAIR*, leading to a changed distal RNA secondary structure and increased *FLC* expression[11,44]. High *FLC* expression and late flowering might therefore be favorable to growth and survival in the harsher winters of N. Sweden, or alternatively, the SSF414N-derived low *FLC* expression and early flowering might favor the relatively mild growth environment in S. Sweden but not the conditions in N. Sweden (Supplementary Fig. 10a, b); therefore, the SSF414N accessions might lack the ability to spread to N. Sweden. However, the distinct distribution could also represent groups from different ice age refugia[45].

A phylogenetic analysis showed that SSF-type genes were present only in dicots, unlike FCA-type genes that existed in dicot and monocot plants. The presence of only one FCA-like gene in *Amborella*, the basal angiosperm, suggested a duplication event during angiosperm evolution. *Aquilegia* represents a phylogenetic midpoint between models such as *Arabidopsis* (core eudicots) and *Oryza* (monocots)[46]. The finding that both SSF and FCA were present in *Aquilegia* and all other dicots, but not monocots, suggested that SSF and FCA-like proteins were remnants of an ancient duplication that occurred after dicot-monocot specialization but before core eudicot/lower eudicot diversification. Alternatively, this event represented a more ancestral duplication, with SSF-type genes being lost in the monocot lineage. Regardless of the origin, SSF genes have contributed to natural variation in flowering in *A. thaliana* through an amino acid polymorphism that influences the protein stability.

In summary, analysis of natural variation in *FLC* regulation followed by molecular dissection of that variation led to the identification of an *FCA* homolog with antagonistic functions. The amino acid polymorphism SSF-N414D can be recognized by the E3 ubiquitination system and contributes to phenotypic evolution. These findings will aid understanding of the functional significance of the amino acid polymorphism important for phenotypic variation in flowering.

## Methods

**Plant materials and cold treatment**. The *Arabidopsis* accessions, including Ull2-5 and Lov-1, were ordered from NASC (http://arabidopsis.info). Seeds were sown in GM-glucose medium and stratified at 5 °C for 3 days. For the nonvernalized sample (NV), seedlings were grown under a long-day (22 °C, 16 h light, and 8 h dark) photoperiod for 14 days; for vernalization, seedlings were grown under standard growth conditions (22 °C, 16 h light and 8 h dark) for 7 days and then transferred to vernalization conditions for a certain duration (5 °C, 8 h light and 16 h dark). After vernalization, we moved the seedlings back to standard growth conditions and harvested the samples 10 and 30 days post-cold treatment (T10 and T30, respectively). The *ssf-1*, *ssf-2*, and *cul1-1* mutants were also ordered from NASC and the homozygous lines were screened with genotyping primers from the TAIR website.

**Population and statistical analysis**. GWAS was performed in the GWA-Portal (gwas.gmi.oeaw.ac.at)[47] using both population structure-corrected (accelerated mixed model-analysis #26447) and -uncorrected (linear model-analysis #28610) methods. Both analyses utilized SNPs from Horton et al.[17] and a minor allele count cutoff of 15. *FLC* expression data is phenotype #42173 in the GWA-Portal. The results of online analysis are open to public access. Boxplots were generated in GraphPad Prism 8.0.0 for windows (GraphPad Software, San Diego, California, USA, www.graphpad.com), and statistical analyses were performed with SPSS version 16 software using the two-tailed unpaired *t*-test method (SPSS Inc., Released 2007, SPSS for Windows, version 16.0. Chicago, SPSS Inc).

**Amino acid sequence comparison**. The predicted SSF (At2g47310) and FCA (At4g16280) amino acid sequences of natural *A. thaliana* accessions were obtained from the 1001 Genomes website (www.1001genomes.org/) and aligned with BioEdit software version 7.0.9.0 using the ClustalW multiple sequence alignment method[48].

**Gene expression analysis**. Total RNA was extracted with the hot phenol extraction method, and all reverse transcription (RT) assays were performed with the Invitrogen Superscript III Reverse Transcription System according to the

manufacturer's instructions. The primers used for qRT-PCR are listed in Supplementary Data 2. The *UBC* gene was used as an internal control.

To evaluate the contribution of the QTL on chromosome 2 (Chr2-QTL) to the *FLC* expression variation in the F2 population (Ull2-5×Lov-1)[8], we first screened 279 F2 lines for common chromosome 5 genotypes, allowing us to isolate the effects of the Chr2-QTL. To achieve this goal, we used the closest molecular marker to each QTL peak to identify lines with heterozygous genotypes at both QTLs on chromosome 5. The resulting 65 lines were genotyped for the Chr2-QTL using the At2g47310 marker and separated into three groups (17 Chr2-QTL/Ull2-5 lines, 19 Chr2-QTL/Lov-1 lines, and 29 Chr2-QTL/Hets). Finally, The RNA of the F2 lines from Chr2-QTL/Ull2-5 and Chr-QTL/Lov-1 groups was extracted separately and applied for *FLC* expression analysis.

A pooling method was used for the *FLC* expression analysis of *SSF* transgenic plants[11,36] (Fig. 3c). For each construct, we generated and randomly selected 50 independent transgenic lines for RNA preparation and pooling. RNA of different transgenic lines was extracted separately and quantified. The same amount of RNA from every transgenic lines of each construct was mixed together to generate a sample pool. In this way, multiple pools were prepared for the transgene of each construct. RT and qPCR were performed using the sample pools, and the statistical analysis showed the difference between the sample pools of different transgenes.

**Site-directed mutagenesis and transformation of SSF variants**. For the construction of SSF-Col, the SSF genomic region from the Columbia accession, containing 1638 bp of the native promoter, all exons and introns, 856 bp of the 3′UTR, and the terminator, was amplified with high-fidelity Taq enzyme using primers plus proper restriction enzyme cut sites: KpnI SSF F1 and SSF NotI R2. The PCR product was cloned into the pGEM-T Easy vector (Promega) and sequenced, and then, the correct SSF fragments were subcloned into the binary vector pCambia 1300. For SSF-ColN414D mutagenesis, the overlap PCR method was applied using the following primers: SSF MfeI F1 with SSF_2578 A-G R1 and SSF_ 2578 A-G F2 with SSF NotI R2 (Supplementary Data 2). Then, the site-mutated fragment was sequenced and subcloned with MfeI and NotI enzyme cleavage sites into pCambia 1300-SSF-Col to obtain pCambia 1300-SSF-ColN414D. Finally, the constructs were transferred into competent *Agrobacterium* cells and transformed into *ssf-2* mutant with the floral dip method.

For the Col-SSF-GFP construct, the SSF genomic region containing the native SSF promoter, exons, and introns was amplified from Col-0 with primers KpnI SSF F1 and SSF SalI R1, the GFP fragment was amplified with primers SalI linker GFP F and BamHI GFP R, and the SSF 3′ UTR and terminator region were amplified with primers BamHI SSF F2 and SSF NotI R2. All the fragments were cloned into pGEM-T Easy, sequenced, and subcloned into the pBluescript vector in that order, and the entire SSF-GFP sequence was cloned into the binary vector pGeen0179 (http://www.pgreen.ac.uk/JIT/pG0179.htm). The construct was then transformed into *Agrobacterium* together with the pSoup plasmid (http://www.pgreen.ac.uk/a_pls_fr.htm) and into the *ssf-2* mutant using the floral dip method.

**Y2H analysis**. The experiment was performed according to the instructions in the Matchmaker Gold Yeast Two-Hybrid user manual from the manufacturer CLONTECH Company (PT4084-1). To compare the interaction between SSF variants and CUL1, we transformed each variant together with CUL1 into yeast cells in parallel and grew the transformants on selection medium containing proper antibiotics without leucine and tryptophan for 3 days. Then, multiple clones were picked and cultured in liquid medium without leucine and tryptophan for 24 h, and the cells were precipitated and washed with liquid culture medium without leucine, tryptophan, adenine, and histidine three times. The yeast concentration was then determined by spectrophotometry, measuring the OD600, and the same amount of yeast cells was diluted in series and spread on solid culture medium without leucine, tryptophan, adenine, and histidine. Clone growth was subsequently compared, and the colonies were photographed. The interaction between FCA and CUL1 was also completed according to the steps described above.

**BiFC assay**. For construction, the SSF coding region was fused in-frame to the N-terminal region of YFP, and the CUL1 coding region was fused in-frame to the C-terminal region of YFP. The correct constructs were cotransformed into wild-type Col-0 protoplasts. After being kept in the dark for 16 h, the cells were imaged using a Zeiss confocal imaging system. The interaction between ZmPYL10 and ZmPP2C14 was used as a positive control[49].

**Homolog identification and phylogenetic tree construction**. The amino acid sequence of *FCA*-like genes and the hidden Markov model (HMM) profile of the RRM_1 domain (Pfam no. PF00076) were first used as queries to perform a tblastn search against all the coding sequences in each genome, with the expectation value (E) set to 1.0. All nonredundant targeted sequences were submitted to Pfam with default settings to confirm the presence of the RRM_1 and WW domains. Only genes encoding proteins harboring both RRM_1 and WW domains were regarded as homologs of *FCA*-like genes.

The maximum-likelihood (ML) method was used to construct a phylogenetic tree according to the complete amino acid sequences of these candidate genes. They were aligned with ClustalW using the default option in MEGA v5.0[50]. Next,

poorly aligned regions with more than 20% gaps or similarity scores lower than 0.001 were removed using TrimAL2.0[51]. We then used Prottest 3.4 to detect the best fit model of protein evolution to construct the ML phylogenetic tree[52]. The best model according to the corrected Akaike information criterion (AICc) was the JTT + I + G + F model (with lnL = 29343.62). We performed this model in PhyML3.1 with 100 bootstrap replicates.

**Genetic distance analysis of FCA and SSF in different species**. The protein sequences of FCA and SSF were obtained as described above, and the genetic distance was calculated with MEGA 5.0 using the Kimura model. The data were incorporated into SPSS 16.0 for normal distribution, *Student's t* test, and boxplot analyses using default parameters.

**Cell-free protein degradation assay**. Total cell extract was obtained from 14-day-old seedlings with extraction buffer (25 mM Tris-MES (pH 7.5), 10 mM NaCl, 10 mM $MgCl_2$, 4 mM PMSF, 5 mM DTT, and 10 mM ATP). Purified and quantified recombinant His-SSF414N or His-SSF414D protein was added to equal amounts of Col-0 or *cul1-1* extract and incubated at 22 °C for different time periods. MG132 was added to the extraction buffer at a concentration of 50 μM for specified samples. The protein was separated by SDS polyacrylamide gel electrophoresis (PAGE) and detected with an anti-His antibody (Abcam, ab184607, dilution: 1:2000).

**Western blot, protein pulldown, and coimmunoprecipitation analysis**. For western blot, total protein was extracted from 14-day-old seedlings with protein extraction buffer (50 mM Tris-HCl (pH 8.0), 150 mM NaCl, 0.1% IGEPAL, 2.5 mM EDTA, 10% glycerol, 10 mM 2-mercaptoethanol MCH, 1 mM PMSF, 10 μM leupeptin, and 1×Roche protease inhibitor cocktail). After incubation on a slowly rotating shaker in a cold room (4 °C) for 10 min, the samples were centrifuged for 15 min at 11,560 × $g$ three times until the supernatant was clear. SDS-PAGE was used to separate the proteins, and the target proteins were detected by immunoblotting with Anti-FCA (FCA(N) antibody) (Abiocode, R3399-1, dilution: 1:2000), or Anti-Ubiquitin antibody [Ubi-1] (Abcam, ab7254, dilution: 1:2000) antibodies.

For pulldown assays, total protein was extracted from tobacco leaves after 3 days of infiltration with the construct containing CUL1-Flag with protein extraction buffer (50 mM Tris-HCl (pH 8.0), 150 mM NaCl, 0.1% IGEPAL, 2.5 mM EDTA pH 8.0, 10% glycerol, 10 mM 2-mercaptoethanol MCH, 1 mM PMSF, 10 μM leupeptin, and 1×Roche protease inhibitor cocktail). After incubation on a slowly rotating shaker in a cold room (4 °C) for 10 min, the samples were centrifuged for 15 min at 11,560 × $g$ three times until the supernatant was clear. Five micrograms of His-SSF fusion protein bound to protein A beads was added to total protein extract samples, incubated overnight at 4 °C, and then washed multiple times with washing buffer (50 mM Tris-HCl (pH 8.0), 150 mM NaCl, 0.1% IGEPAL, 2.5 mM EDTA pH 8.0, 10% glycerol, 1 mM PMSF). SDS-PAGE was used to separate the proteins, and the target proteins were detected by immunoblotting with anti-Flag (MBL, M185, dilution: 1:10,000) or anti-His (Abcam, ab184607, dilution: 1:2000) antibodies.

For Co-IP assay, the SSF414N/414D-GFP and CUL1-Flag vectors were cotransformed into wild-type Col-0 protoplasts. After being kept in the dark for 16 h, total protein was extracted with protein extraction buffer (50 mM Tris-HCl (pH 8.0), 150 mM NaCl, 0.1% IGEPAL, 2.5 mM EDTA, 10% glycerol, 10 mM 2-mercaptoethanol MCH, 1 mM PMSF, 10 μM leupeptin, and 1×Roche protease inhibitor cocktail) and incubated with GFP-Trap beads (ChromoTek, gtma-20) for 2 h. The immunoprecipitated samples were washed five times with extraction buffer, separated by SDS-PAGE, and subjected to immunoblot analysis with anti-GFP (Roche, 11814460001, dilution: 1:1000) or anti-Flag (MBL, M185, dilution: 1:10,000) antibodies. The antibodies were validated by the manufacturer.

**Mass spectrometry analysis**. SDS-PAGE gel pieces were destained and then incubated with trypsin (Promega, 10 ng/μL) overnight at 37 °C. Peptides were extracted by incubation with 5% TFA for 1 h, followed by the addition of 2.5% TFA/50% ACN for 1 h at 37 °C. The combined supernatants were dried in a SpeedVac concentrator (Thermo Fisher) for MS analysis. Samples were analyzed by nanoLC-MS/MS using an RSLC system interfaced with a Q Exactive instrument (Thermo Fisher, San Jose, CA). Mass spectrometry data were acquired using a data-dependent acquisition procedure with a cyclic series of a full scan acquired with a resolution of 120,000 followed by MS/MS scans (30% collision energy in the HCD cell) with a resolution of 30,000 for the 20 most intense ions with a dynamic exclusion duration of 20 s. The LC-MS/MS peak list files from each experiment were searched against the corresponding TAIR10 database (35,386 sequences) using an in-house version of X! tandem (SLEDGEHAMMER (2015.09.01), thegpm.org) with carbamidomethylation on cysteine as a fixed modification and oxidation of methionine as a variable modification. Values of +/−7 ppm and 20 ppm were used as tolerances for precursor and product ions, respectively. The false discovery rate was estimated for all samples by using a reverse database (FDR < 0.01). All the identified spectra fit the following criteria: the peptide is present in the *Arabidopsis* database, peptide log(E) ≤ −2 (E value ≤ 0.01).

**Transient gene expression analysis in tobacco leaves**. The *FLC* promoter was cloned into a pGreenII 0800-LUC vector to generate *FLC: LUC*, and *SSF* was cloned into pGreenII 62-SK for *SSF* overexpression. Different combinations of vectors were infiltrated into the leaves of *N. benthamiana* plants. Two days after infiltration, leaves were collected to capture the LUC activity using a NightShade LB 985 Plant in vivo Imaging System. Total proteins were extracted with extraction buffer (50 mM Tris-HCl (pH 8.0), 150 mM NaCl, 0.1% IGEPAL, 2.5 mM EDTA pH 8.0, 10% glycerol, 10 mM 2-mercaptoethanol MCH, 1 mM PMSF, 10 μM leupeptin, and 1×Roche protease inhibitor cocktail). Then, the activities of the firefly (*Photinus pyralis*) and Renilla (*Renilla reniformis* or sea pansy) luciferases were measured using a Dual-Luciferase Reporter Assay System (Promega, E1910) through a TriStar² LB 942-Multimode Microplate Reader.

**ChIP assays**. RNA Pol II and SSF-GFP ChIP on *FLC* were conducted. Briefly, 3 g of 12-day-old seedlings was cross-linked with 1% formaldehyde, and the nuclei were extracted and sonicated. After the chromatin was precleared with protein A/G magnetic beads, the protein-DNA complexes were precipitated with anti-GFP (Ab290, Abcam) or anti-Pol II (8WG16, ab817, Abcam) antibodies. The protein-DNA complexes were reverse cross-linked overnight with 0.2 M NaCl at 65 °C. The DNA was extracted and amplified by qPCR. The gene-specific ChIP primers are shown in Supplementary Data 2. Antibody amounts were 15 μg for anti-GFP and 5 μg for anti-Pol II.

**Reporting summary**. Further information on research design is available in the Nature Research Reporting Summary linked to this article.

## Data availability

Data supporting the findings of this work are available within the paper and its Supplementary Information files. A reporting summary for this Article is available as a Supplementary Information file. The datasets and plant materials generated and analyzed during the current study are available from the corresponding author upon request. Source data are provided with this paper.

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

## Acknowledgements

We thank Professor Caroline Dean (C.D.), Dr. Xiaofeng Fang, and the C.D. laboratory members from John Innes Centre for technical assistance and useful discussions, Professor Chuanxiao Xie from the Chinese Academy of Agricultural Sciences for providing the BiFC vectors, and Dr. Weina Si from Anhui Agricultural University for helping with the SSF phylogenetic analysis. This research was supported by The National Key Research and Development Program of China (2016YFD0101803, 2017YFD0301301), National Natural Science Foundation of China (31670264), UK Biotechnology and Biological Sciences Research Council (BBSRC) grant BB/I007857/1, and Institute Strategic Program grant BB/J004588/1.

## Author contributions

P.L., Y.W., and Z.T. designed the research. Y.W., Z.T., and J.S. performed the gene expression assay. D.F. performed the GWAS of *FLC* expression. Y.W., Z.T., C.Q., Y.Z., H.W., and S.R. performed the vector construction and analysis of transgenic plants. Y.W., W.W., and H.W. performed the yeast two-hybrid assays and protein analysis. Y.W., W.W., and C.W. performed the BiFC protein expression and confocal assays. P.L., Y.W., and Z.T. wrote the article.

## Competing interests

The authors declare no competing interests.
