## [Peer Review File · Nature Communications]

Reviewers' comments:

Reviewer #1 (Remarks to the Author):

The authors identify a naturally occurring allele of an FCA homologue, *SSF*, in *Arabidopsis thaliana* that causes a late flowering phenotype. The authors provide evidence that plants harboring the variant that causes late flowering (*SSF-D*) express the flowering time repressor *FLC* at a higher level than plants that harbor the second variant (*SSF-N*), which flowers comparatively early. The authors provide evidence that the *SSF-D* variant is causal for the late flowering of specific analyzed accessions of *A. thaliana* and that *FLC* levels correlate with expression levels of *SSF*.

In a *Col-0* background, *ssf* mutants flower only 2-4 leaves later in LDs and only 4 leaves later in *Col-0*, making it a relatively mild flowering time regulator in this background. This is in spite of *FLC* levels being reduced to 15-20% of normal levels, although this is in a background that does not contain the *FLC* activator *FRI*. In this background, inactivation of *ssf* has a very variable effect on flowering time (although the authors unjustly claimed that these plants were earlier flowering, an experiment with a higher number of plants is required here) and in keeping with this, *FLC* levels are only mildly reduced. When transgenic plants are generated that contain the *Col-0* *SSF-N* version as well as the *SSF-D* version, the latter genotypes flowered with many more leaves and they expressed *FLC* at higher levels, which supports the conclusions from the authors that *SSF-D* natural variants delay flowering time.

The authors suggest that the basis for this phenotype is a disrupted interaction with the 26S proteasome and therefore a longer half-life of the *SSF-D* protein relative to the *SSF-N* form. This longer half-life would then result in higher levels of *FLC* expression as the authors provide evidence that *SSF* directly regulates *FLC* expression.

While the topic is an important one and the authors have performed a reasonably good job at dissecting the nature of *SSF* activity, the fact that the *SSF-D* allele is randomly distributed across Spain and Germany comes as a surprise and the authors do not address this adequately. Furthermore, there are quite a few technical issues with the manuscript that need to be addressed to provide more robustness for the claims of the authors.

Specific comments:

- Flowering time is not well assayed the paper in general. There is a mixture of leaf counting and days to bolting. It would be preferable to see both metrics to adequately judge the manuscript.
- Student's t-test alone is not a sufficient description. In addition, the number of plants counted in each experiment is missing
- The PolII experiments are not convincing as the regions with the lowest WT signal are the only regions with depleted PolII in *ssf-2*
- The ChIP experiments are also unconvincing since the enrichment is quite low and there is no discernable pattern of enrichment (i.e. why is the -253 position not slightly enriched since chromatin shearing is unlikely to be able to discriminate so easily from -501 to -253) - to address this I suggest the authors design primers that flank enriched regions to show a peak of enrichment and/or provide more evidence that *SSF* binds to the *FLC* region.
- The experiments with transgenic *SSF-N* and *SSF-D* are very nice but it would be more convincing to select stable lines that express each transgene at similar levels and to assay the flowering time behavior
- I do not find the BiFC experiments convincing, the authors should repeat them and select better images. Controls would also be useful (i.e. a protein you do not expect to interact and a protein you do expect to interact)
- Controls for the protoplast experiments would also be desirable (e.g. a UBQ10-mCherry in the background as a normalization signal). The same goes for analyzing the expression in the roots of *SSF-GFP* plants.

- In many of the legend it is written that the experiment was performed three times. Does this mean biological or technical replicates. Indeed, many experiments appear to have been performed only once

Reviewer #2 (Remarks to the Author):

Review of Wang et al for Nat.Comm.

In this manuscript the authors investigate the cause of natural variation in flowering time in Arabidopsis. They use a clever GWAS approach to identify a novel regulator of the major flowering time gene FLC. They move from GWAS and QTL studies to knockouts and transgenics to show that allelic variation in their candidate gene SSF does cause variation in flowering time and FLC regulation. They further investigate the mechanistic basis for the allelic variation: the allelic variation affects SSF's binding to CUL1 and therefore SSF stability. This is an impressive study that goes all the way from GWAS to functional analysis of allelic variants. There are a few minor details that need attention, as described below:

1) The paper needs substantial editing to correct numerous typos, grammatical errors, etc. If Nature Communications does not provide substantial copy editing in house, then the authors need to engage a professional editing service (and based on my experience I recommend that they choose a service based in the US or UK).

2) In some figures the error bars represent standard deviation and in others standard error. Example, Fig. 2 b-h vs I,j. Is there a justification for this? If so please explain. Otherwise choose one and stick with it. Figures 5 and 6 do not indicate if error bars are SEM or SD

3) In many cases the number of samples or replicates is not provided. For example, Fig 1c,d; Fig 2; Fig 3c (how many independent pools?); and maybe others.

4) Line 78 about E3 involvement in natural variation is incorrect; see <https://pubmed.ncbi.nlm.nih.gov/17417637-a-qt1-for-rice-grain-width-and-weight-encodes-a-previously-unknown-ring-type-e3-ubiquitin-ligase/> AND <https://pubmed.ncbi.nlm.nih.gov/32131496-a-ring-type-e3-ubiquitin-ligase-osgw2-controls-chlorophyll-content-and-dark-induced-senescence-in-rice/>

5) Line 34 "evolutionary importance" has not been demonstrated.

Reviewer #3 (Remarks to the Author):

At2g4710 (Sister of FCA/SSF) contributes to natural variation in flowering time of Arabidopsis accessions. SSF occurs in two forms. SSF414N or SSF414D. SSF414N interaction with CUL1-E3 is stronger than SSF114D interaction with CUL1-E3 that affects SSF protein degradation differentially, which in turn modulates FLC expression.

This paper is an interesting paper. I find the data presented in this paper conclusively shows few things. 1) Mutations in the SSF genes appear to show early flowering. 2) SSF interacts with CUL1. 3) The polymorphism in SSF1 (N414D) potentially impacts the degradation of SSF. However, the links between all three of these is not really well established and the paper suffers from several deficiencies as I describe below.

Major concerns:

I am of the view that their GWAS analysis is not very informative. I do not believe the signals they see (i.e., 30 regions associated with vernalization response) are actually reflective of the biology. They appear to me as background noise given the significance levels of many of these SNPs are almost identical. The idea of focusing on one region, which in itself does not appear to provide any significant clean signal, is not at all justified. While their analysis may be true, I do not believe there is a shred of evidence to focus on SSF from the GWAS data presented in this paper.

Fig 1B is a reproduction of part of their previously published figure (Fig. 2D of Li et al, *Genes and Development*, 2014). But it is not explicitly stated. The methods and everything is described without proper citation of their previous work. This may be an oversight. But nevertheless, unacceptable without properly citing/explicitly stating. Also, their paper in 2014 showed that in this cross, there are peaks in Chr 5 near top of chr 5 and bottom of Chr 5. How did the authors control for the effect of these QTLs when they pooled the F2 population (6 plants each) on the basis of the marker genotype at Chr 2? It is very misleading to just show Chr 2 and then present it as if it is the major effect QTL.

Supplementary Fig 3. Please show a schematic of the gene and the position of LP and RP. The position of insertions are shown in Fig. 2a. But it would be better here to show the position of primers.

Fig. 3 b. The authors should include the distribution of the flowering times of *ssf-2* itself as a control in this figure. This lacks statistical support. It is unclear to me as to what is meant by "50 independent T2 transgenic lines from each construct were mixed to make each sample".. They report they had 72 and 63 lines respectively for SSF414N and SSF414D. However, there are roughly 10+ data points in each.. Please explain? There are two data points that appear to be outliers (high FLC expression).. Do these two datapoints drive the statistical significance?

Do the SSF-GFP constructs rescue *ssf2* mutant?

Supplementary Fig 4 : No statistics

Supplementary Fig 5: No statistics

Supplementary Figures 8 & 9. What is the point of showing flowering time differences between Northern/Southern Sweden accessions, while the grouping can be done through the genotypes of the SSF414N vs SSF414D?

Reviewers' comments:

Reviewer #1 (Remarks to the Author):

The authors identify a naturally occurring allele of an FCA homologue, SSF, in *Arabidopsis thaliana* that causes a late flowering phenotype. The authors provide evidence that plants harboring the variant that causes late flowering (SSF-D) express the flowering time repressor FLC at a higher level than plants that harbor the second variant (SSF-N), which flowers comparatively early. The authors provide evidence that the SSF-D variant is causal for the late flowering of specific analyzed accessions of *A. thaliana* and that FLC levels correlate with expression levels of SSF.

In a Col-0 background, *ssf* mutants flower only 2-4 leaves later in LDs and only 4 leaves later in Col-0, making it a relatively mild flowering time regulator in this background. This is in spite of FLC levels being reduced to 15-20% of normal levels, although this is in a background that does not contain the FLC activator FRI. In this background, inactivation of *ssf* has a very variable effect on flowering time (although the authors unjustly claimed that these plants were earlier flowering, an experiment with a higher number of plants is required here) and in keeping with this, FLC levels are only mildly reduced. When transgenic plants are generated that contain the Col-0 SSF-N version as well as the SSF-D version, the latter genotypes flowered with many more leaves and they expressed FLC at higher levels, which supports the conclusions from the authors that SSF-D natural variants delay flowering time.

Response: Thank you for the Reviewer's comments. To further verify the early flowering phenotype of the *ssf* mutant, we assayed the flowering time again with a large number of plants, namely, 48 mutant and 48 wild-type Col-0 individuals. This finding showed that the bolting time (days to bolting) of the *ssf* mutant was approximately one week earlier than that of Col-0 under long-day growth conditions and approximately 10 days earlier under short-growth conditions (Fig. 2b, c, Supplementary Fig. 3b, d). These data were consistent with the leaf number change (Supplementary Fig. 3c, e). In addition, in the presence of the *FLC* activator FRI, the *SSF* mutation also significantly caused early flowering (13 leaf difference, Student's t test, $p=0.0032$, Supplementary Fig. 3f).

We agree with the reviewers and used bolting time to show the flowering time in the revised manuscript; we also placed the total leaf number data in the Supplementary figures.

The authors suggest that the basis for this phenotype is a disrupted interaction with the 26S proteasome and therefore a longer half-life of the SSF-D protein relative to the SSF-N form. This longer half-life would then result in higher levels

of *FLC* expression as the authors provide evidence that *SSF* directly regulates *FLC* expression.

While the topic is an important one and the authors have performed a reasonably good job at dissecting the nature of *SSF* activity, the fact that the *SSF-D* allele is randomly distributed across Spain and Germany comes as a surprise and the authors do not address this adequately. Furthermore, there are quite a few technical issues with the manuscript that need to be addressed to provide more robustness for the claims of the authors.

Response: Yes, we agree with the reviewer that we only showed the geographical distributions across the three countries in the previous version but did not provide a sufficient explanation as required. To explore the possible reason why the *SSF-D* alleles are randomly distributed across Spain and Germany while the *N* form exists predominantly in S. Sweden and the *D* form is mainly observed in N. Sweden, we collected and compared the highest and lowest temperatures on average between the north and south areas of Sweden (Göteborg versus Kiruna), Spain (Málaga versus Bilbao) and Germany (München versus Hamburg). The findings showed that in Spain and Germany, the monthly temperature difference between the northern and southern areas is quite small and ranges from 2-5 degrees, while in Sweden, the monthly temperature difference between the northern and southern areas is much larger and ranges from 4 to 16 degrees (Supplementary Fig. 9b).

Temperature has been considered one of the most important factors affecting plant flowering (Duncan et al, *Elife*, 2015, 10.7554/eLife.06620), and we reason that the large temperature difference in Sweden may exert great selection pressure on plant adaptation. The *SSF414D*-derived higher *FLC* expression and later flowering may be more favorable for the lower temperatures in N. Sweden, whereas the *SSF414N*-derived lower *FLC* expression and early flowering may be more favorable for the mild climate in S. Sweden (Fig. 3b-f, Supplementary Fig. 9a, b). The temperature in the northern and southern areas of Spain and Germany is similar and may have a similar influence on the plants in these regions, which could be one of the reasons that *SSF414N* and *SSF414D* are distributed quite randomly across Spain and Germany but not in Sweden as shown in Fig. 7a. However, we cannot exclude the possibility that the geographical distribution in these three countries may be associated with refugia after the last ice age. We included these analyses in Supplemental Fig. 9 and revised the related discussion section.

To fully understand the role of *SSF* variants in adaptation, the transgenic plants containing the *N/D* variants under the same genetic background need to be grown in the natural field environment in Sweden for the comparison of phenotypes and plant survival in the future.

Specific comments:

- Flowering time is not well assayed the paper in general. There is a mixture of leaf counting and days to bolting. It would be preferable to see both metrics to

adequately judge the manuscript.

Response: In the study of *Arabidopsis thaliana*, two kinds of data are normally used to assay flowering time, i.e., bolting time or total leaf number, and these two types of data are usually consistent with each other (M. Koornneef et al, Mol Gen Genet, 1991, doi: 10.1007/BF00264213; B. Mendez-Vigo et al, Journal of Experimental Botany, 2010, doi: 10.1093/jxb/erq032). Therefore, in our previous manuscript, to reduce labor work for flowering time assays, we chose to use the bolting time method for a large number of plants, such as for the F2 population, which has many individuals, and very late flowering plants, which have a high number of leaves. However, for early flowering plants with a small number of leaves, we chose to use the total leaf number instead.

We fully agree with the reviewer's comment. To make the data more consistent across the paper, we performed more experiments to assay the bolting time to replace the old total leaf number data in the main figures and moved the total leaf number data to supplementary figures based on the reviewer's suggestion. Consistent with the findings in literatures, we also found that the bolting time data were consistent with the total leaf number data (Fig. 2b, 2c, 3f, 5g, Supplementary Fig. 3b-f, 4a, b, 9a).

- Student's t-test alone is not a sufficient description. In addition, the number of plants counted in each experiment is missing.

Response: We agree with the reviewer and added more details on the analysis methods, such as significant p-values based on Student's t test and the number of samples for all experiments; moreover, all the legends have also been updated.

- The PolIII experiments are not convincing as the regions with the lowest WT signal are the only regions with depleted PolIII in *ssf-2*

Response: We are grateful for the reviewer's comments, which have been very helpful for improving the data accuracy. We repeated the Pol II ChIP experiment several more times with the improved ChIP protocol modified at the chromatin sonication step and obtained better results, which showed higher Pol II enrichment. In the *ssf-2* mutant, the Pol II enrichment was significantly reduced in the promoter, gene body and 3'UTR of the *FLC* genome region (Figure 2i, k).

- The ChIP experiments are also unconvincing since the enrichment is quite low and there is no discernable pattern of enrichment (i.e. why is the -253 position not slightly enriched since chromatin shearing is unlikely to be able to discriminate so easily from -501 to -253) - to address this I suggest the authors design primers that flank enriched regions to show a peak of enrichment and/or provide more evidence that SSF binds to the FLC region.

Response: Thank you for these valuable comments. We performed additional ChIP experiment using the SSF-GFP transgenic line, which fully rescued the *ssf-2* flowering time (Supplementary Fig. 4a, b), based on the improved ChIP protocol modified at the sonication step. In addition, we used more qPCR primers. The results showed that SSF occupies the promoter, gene body and 3'UTR of the *FLC* genomic region (Fig. 2j, k).

- The experiments with transgenic SSF-N and SSF-D are very nice but it would be more convincing to select stable lines that express each transgene at similar levels and to assay the flowering time behavior

Response: We agree and selected the SSF414N and SSF414D transgenic lines that have similar SSF expression levels for flowering time comparisons (Fig. 3d), and the results showed that the SSF414D plants are later flowering than the SSF414N plants (Fig. 3f). Consistent with the flowering time change, the SSF414D plants have higher *FLC* expression than the SSF414N plants (Fig. 3e).

- I do not find the BiFC experiments convincing, the authors should repeat them and select better images. Controls would also be useful (i. e. a protein you do not expect to interact and a protein you do expect to interact)

Response: Thank you for these comments. Accordingly, we performed additional BiFC experiments and obtained better images. In the revised manuscript, the interaction of ZmPYL10 and ZmPP2C14, which has been published (Wang et al, Plant Molecular Biology, 2018, doi: 10.1007/s11103-017-0692-7), was used as a positive control and more negative controls were included to make sure our data are reliable. The results are shown in Fig. 4b.

- Controls for the protoplast experiments would also be desirable (e.g. a UBQ10-mCherry in the background as a normalization signal). The same goes for analyzing the expression in the roots of SSF-GFP plants.

Response: Yes, we agree with this comment. The reviewer's suggestion is important to make our data more accurate. Because the efficiency of the transformation of multiple plasmids into the protoplast is not sufficiently high to obtain abundant cotransformed cells that are required for data quantification, so instead of UBQ10-mCherry, we used chlorophyll fluorescence as an internal control for the normalization of the GFP signal in the transformed protoplast cells. The new results with normalization are consistent with our previous data (Fig. 5a-d).

Similarly, for the analysis of GFP in roots, we used DAPI-stained nuclei as an internal control to normalize the GFP signal in transgenic plants, and a consistent conclusion was obtained. These new results are shown in Fig. 6c, d.

- In many of the legend it is written that the experiment was performed three times. Does this mean biological or technical replicates. Indeed, many experiments appear to have been performed only once

Response: We apologize for not describing the legends clearly. All the results in the figures represent the means of more than three biological replicates. Accordingly, in the revised version, we have added a more detailed description in each legend, such as the number of biological replicates and how many individual plants were used in the assays.

Reviewer #2 (Remarks to the Author):

Review of Wang et al for Nat.Comm.

In this manuscript the authors investigate the cause of natural variation in flowering time in Arabidopsis. They use a clever GWAS approach to identify a novel regulator of the major flowering time gene FLC. They move from GWAS and QTL studies to knockouts and transgenics to show that allelic variation in their candidate gene SSF does cause variation in flowering time and FLC regulation. They further investigate the mechanistic basis for the allelic variation: the allelic variation affects SSF's binding to CUL1 and therefore SSF stability. This is an impressive study that goes all the way from GWAS to functional analysis of allelic variants. There are a few minor details that need attention, as described below:

1) The paper needs substantial editing to correct numerous typos, grammatical errors, etc. If Nature Communications does not provide substantial copy editing in house, then the authors need to engage a professional editing service (and based on my experience I recommend that they choose a service based in the US or UK).

Response: Thank you for these valuable comments, which we agree with. We have hired the professional English editing service America Journal Experts (AJE, www.aje.com), which has substantially improved our manuscript.

2) In some figures the error bars represent standard deviation and in others standard error. Example, Fig. 2 b-h vs I, j. Is there a justification for this? If so please explain. Otherwise choose one and stick with it. Figures 5 and 6 do not indicate if error bars are SEM or SD

Response: There is no justification for the error bars. We agree with the comment and have changed SD to SE throughout the whole revised version and add the SE to the legends of Figure 5 and 6. Thank you.

3) In many cases the number of samples or replicates is not provided. For example, Fig 1c,d; Fig 2; Fig 3c (how many independent pools?); and maybe others.

Response: We apologize for our negligence. We agree and have added the sample number in the revised legends for all figures.

4) Line 78 about E3 involvement in natural variation is incorrect; see <https://pubmed.ncbi.nlm.nih.gov/17417637-a-qt1-for-rice-grain-width-and-weight-encodes-a-previously-unknown-ring-type-e3-ubiquitin-ligase/>

AND

<https://pubmed.ncbi.nlm.nih.gov/32131496-a-ring-type-e3-ubiquitin-ligase-osgw2-controls-chlorophyll-content-and-dark-induced-senescence-in-rice/>

Response: Thank you. Our description is not appropriate, and we have removed this sentence accordingly.

5) Line 34 “evolutionary importance” has not been demonstrated.

Response: We agree with the reviewer’s comments. The evolutionary importance is always a complex issue for natural variations. Indeed we have not addressed this point enough in the paper. We have deleted this wording accordingly and added a discussion on the possible role of SSF variants on plant adaptation in the discussion section.

Reviewer #3 (Remarks to the Author):

At2g4710 (Sister of FCA/SSF) contributes to natural variation in flowering time of Arabidopsis accessions. SSF occurs in two forms. SSF414N or SSF414D. SSF414N interaction with CUL1-E3 is stronger than SSF114D interaction with CUL1-E3 that affects SSF protein degradation differentially, which in turn modulates FLC expression.

This paper is an interesting paper. I find the data presented in this paper conclusively shows few things. 1) Mutations in the SSF genes appear to show early flowering. 2) SSF interacts with CUL1. 3) The polymorphism in SSF1 (N414D) potentially impacts the degradation of SSF. However, the links between all three of these is not really well established and the paper suffers from several deficiencies as I describe below.

Major concerns:

I am of the view that there GWAS analysis not very informative. I do not believe the signals they see (i.e., 30 regions are associated with vernalization response)

are actually reflective of the biology. They appear to me as background noise given the significance levels of many of these SNPs are almost identical. The idea of focusing on one region, which in itself does not appear to provide any significant clean signal is not at all justified. While their analysis may be true, I do not believe there is a shred of evidence to focus on SSF from the GWAS data presented in this paper.

Response: Thank you for this valuable comment. We understand the reviewer's skepticism about this Manhattan plot and the underlying GWAS. We have repeated the GWAS with a more stringent minor allele count cutoff ($MAC > 15$), which greatly reduces the background signal noted by the reviewer, whereas the peak around At2g47310 remains (Fig. 1a). As for the number of peaks in the Manhattan plot, this is typical of a complex trait, in which variants at many genes contribute to the observed phenotypic variation. While the peak in question is only marginally significant in the GWAS, the significance level is not the only factor taken into account when choosing to explore a GWAS peak. In this case, we provided supporting information (QTLs, gene annotations, etc.) that also played into our decision to pursue At2g47310 as a candidate. The manuscript's molecular analyses of the natural polymorphisms in SSF are also consistent with the peak being "real". We agree with the reviewer that the GWAS itself does not prove that At2g47310 underlies vernalization variation (because GWASs are only statistical associations); however, it was part of the information that provided a starting point for molecular identification and verification of natural genetic variation in the gene.

Fig 1B is a reproduction of part of their previously published figure (Fig. 2D of Li et al, Genes and Development, 2014). But it is not explicitly stated. The methods and everything is described without proper citation of their previous work. This may be an oversight. But nevertheless, unacceptable without properly citing/explicitly stating. Also, their paper in 2014 showed that in this cross, there are peaks in Chr 5 near top of chr 5 and bottom of Chr 5. How did the authors control for the effect of these QTLs when they pooled the F2 population (6 plants each) on the basis of the marker genotype at Chr 2? It is very misleading to just show Chr 2 and then present it as if it is the major effect QTL.

Response: Sorry, the citation of our previous publication is not clear as required. We have added additional citations to the text and additional descriptions on this point in the method section.

For the QTL results, we apologize for not presenting the results in a proper way. The SSF region is the only main QTL on chromosome 2, and there are two QTLs on chromosome 5. We have added a more detailed description in the text to clarify this information and included the QTL results of other chromosomes in Supplementary Fig. 1b together with the citation of our previous paper (Li et al, G&D, 2014).

To determine the contribution of the QTL on chromosome 2 to *FLC* regulation, we used the pooling method as previously reported (V. Coustham et al, Science, 2012, doi: 10.1126/science.1221881; P Li et al, Genes & Development, 2015, doi: 10.1101/gad.258814.115). In brief, we genotyped the F2 population with the marker close to the peak of the QTL on chromosome 2 and divided the F2 into two groups for pooling; in this way, the other chromosome regions, including the QTLs on chromosome 5, will be averaged into the two groups evenly as the segregation of genetic loci are random. Therefore, the difference between the two pools will represent the effect from the QTL on chromosome 2. We have added a more detailed description of the pooling method in the revised manuscript.

Supplementary Fig 3. Please show a schematic of the gene and the position of LP and RP. The position of insertions are shown in Fig. 2a. But it would be better here to show the position of primers.

Response: We agree and have added the schematic of the gene and the position of LP and RP primers for the genotyping of *ssf-1* and *ssf-2* in Supplementary Figure 3a. In addition, we also showed the gene structure of the *CUL1* gene and the primer position of the *cull-1* mutant in Supplementary Figure 5a.

Fig. 3 b. The authors should include the distribution of the flowering times of *ssf-2* itself as a control in this figure. This lacks statistical support. It is unclear to me as to what is meant by “50 independent T2 transgenic lines from each construct were mixed to make each sample” .. They report they had 72 and 63 lines respectively for SSF414N and SSF414D. However, there are roughly 10+ data points in each.. Please explain? There are two data points that appear to be outliers (high FLC expression).. Do these two data points drive the statistical significance?

Response: Thank you for these valuable comments. We have added the flowering time data for the *ssf-2* mutant in Supplementary Fig. 3b as a comparison. The statistical analysis and the significant p-value were added in the figure (Fig. 3b).

We apologize that the pooling method was not described clearly in our previous version. In total, we generated 72 and 63 independent T2 transgenic lines for SSF414N and SSF414D and randomly selected 50 lines from each type of transgene to make each sample pool. In detail, for each type of transgene, we grew the 50 lines and extracted their RNA separately and then the same amount of RNA from each line was measured and mixed together as a sample pool. In this way, 9 pools were generated for SSF414N and 12 pools were generated for SSF414D. We incorporated the method for pooling in the method section and included the sample size in the legends (Fig. 5d).

We apologize for the editing error in our previous version. The two data points that appear to be outliers were from an incorrect copy during figure preparation. In fact, there was only one outlier in the original data; therefore, we updated the figure with the correct one (Fig. 3c, Supplementary Table 2). We performed a significance

test without this outlier and found that the expression difference between SSF414N and SSF414D was even more significant ($p=8.8201E-5$) than that with all of the individual data ($p=0.0032$); therefore, the statistical significance was not driven by this outlier.

Do the SSF-GFP constructs rescue *ssf2* mutant?

Response: Yes, we assayed the flowering time of the SSF-GFP transgenic line and found that SSF-GFP can rescue the *ssf-2* phenotype (Supplementary Fig. 4a, b).

Supplementary Fig 4 : No statistics

Response: We apologize for our negligence. We have added the statistical test results in the figures (Supplementary Fig. 4a-e).

Supplementary Fig 5: No statistics

Response: We have added the statistical test results in the figures (Supplementary Fig. 5c-g).

Supplementary Figures 8 & 9. What is the point of showing flowering time differences between Northern/Southern Sweden accessions, while the grouping can be done through the genotypes of the SSF414N vs SSF414D?

Response: We agree with the reviewer. Accordingly, we compared the flowering time of the accessions in Sweden between the two groups with SSF414N and SSF414D and found that the accessions with SSF414N are significantly earlier flowering than those with SSF414D. We have removed the old figure and replaced it with a new one (Supplementary Fig. 9a).

Reviewer #1 (Remarks to the Author):

The authors have addressed many of my comments adequately from the previous review and the new version of the manuscript is an excellent piece of work. The ChIP-qPCR data for both PolII and SSF-GFP are much more convincing, controls for the BiFC have been included, and the normalization has been applied when quantifying the fluorescent images. I have some comments below that can be addressed with a modification of the text.

In addition, a more thorough analysis of the flowering behaviors of the various mutants and transgenic lines has been done. However, the new flowering time data has some inconsistencies that interfere with the rather singular idea the authors have of SSF stability influencing FLC expression. In LDs, the days to bolting and the leaf numbers match quite well and with published data of *flc-3* mutants. However, in SDs, *ssf* flowers much earlier in terms of days to bolting (20-25 days) whereas the leaf numbers are quite similar to Col-0 (4 leaves). In the literature, *flc-3* mutants tend to flower 7-17 leaves and 8 days earlier in SDs compared to Col-0. Therefore, the leaf initiation rate in SDs appears to be much higher in *ssf*. The summary statement "both mutants showed early flowering and decreased FLC expression in both long-day and short-day growth conditions" (Line 135-136) ignores the differences between leaf numbers and days, which should be acknowledged. Furthermore, the authors do not assay FLC levels in SDs, which is suggested in the summary sentence. This should also be changed to accurately reflect the results shown.

These phenotypes in SDs suggest that there is much more going on in the *ssf* mutant than modulating FLC expression alone. Although the authors acknowledge this, I would suggest a more explicit statement in the abstract that highlights the putative complex nature of the molecular mechanism underlying the *ssf* phenotype and directs the reader to the identification of a specific example of how modifications of SSF activity have been selected during evolution to modify FLC levels in LDs and possibly SDs.

A comment that has not been sufficiently addressed is the statistical analysis. A student's t-test can come in many forms. In the methods, an explicit statement of whether a one-tailed, two-tailed, paired or unpaired test was performed is essential to evaluate the appropriateness of the test.

Reviewer #2 (Remarks to the Author):

My concerns have been adequately addressed in the revision.

Reviewer #3 (Remarks to the Author):

I read with interest the revised version of the manuscript from Peijin Li's group. Just to re-iterate, I had in my earlier review indicated that the manuscript clearly shows 1) Mutations in SSF genes appear to cause early flowering. 2) SSF interacts with CUL1 and 3) The polymorphism in SSF1 (N414D) potentially impacts the degradation of SSF1. The authors now also show the SSF1 polymorphism (N414D) in natural condition is associated with a difference in flowering time (Supp Fig 9a).

I had raised two major concerns earlier. First, I argued that there is no evidence in the GWAS analysis to focus on SSF1. The authors now respond saying that if they increase the minor allele frequency to 15% (i.e., minor allele count of 15 out of 102 lines) in rebuttal it says 15%, in the paper it says 18% the all other noise disappears but the peak around SSF1 remains (Fig 1a). I can't see it in their Figure whether this is significant at all. I do not have a problem in their SSF1 analysis. But arguing that it came through GWAS is in my view simply spinning the story and not really supported by evidence. I am also not sure why they have to justify using this approach. Rather, simply state that we looked for

regions in the GWAS that are potentially harbouring candidate genes and state while there was not a significant peak, there was a potential association (I guess it will turn out to be significant in a simple linear reg, without correcting for pop structure) around SSF1, one of the FCA orthologues. This would be a more accurate description than their story spinning, which I do not support.

My second concern was that they are "reproducing" a figure from their previous publication without properly citing it. Even now they "reproducing" a figure calling it "reanalysing"... There is no reanalysis. Fig 1b and Supplementary Fig 1b and direct "reproduction" of the Figure published by them in *Genes and Development* (Li et al, *Genes and Dev*, 2014, Fig 2D, panel titled ULL2-5 x Lov-1 SV2 x SV1). I also do not understand, why they simply do not state this explicitly that it is a reproduction of a previously published data. The authors simply stat that this QTL has not been studied in depth previously and cite their previous work, while describing all the methods as if it is a new analysis, which is simply not true. I find this very disturbing.

I also disagree with their assertion that if you group your plants on the basis of Chr2 that the chr 5 QTL alleles will be randomly distributed. This is not correct. Typically, ghost QTLs appear when you do not correct for the effect of one QTL. A proper way of doing the experiment would be to genotype the same plants for the Chr 5 QTL and correct for that variation and then ask whether the Chr 2 allelic difference remains. (which it might).. But shrugging it off saying that they will be randomly distributed is not accurate. The key question is whether the expression difference observed is primarily driven by Chr 2 allele or by the LD or any non-random distribution of the Chr 5 alleles given they explain most of the variation, while Chr 2 allele in this cross primarily accounts for only 10% of the variation.

REVIEWER COMMENTS

Reviewer #1 (Remarks to the Author):

The authors have addressed many of my comments adequately from the previous review and the new version of the manuscript is an excellent piece of work. The ChIP-qPCR data for both PolII and SSF-GFP are much more convincing, controls for the BiFC have been included, and the normalization has been applied when quantifying the fluorescent images. I have some comments below that can be addressed with a modification of the text.

In addition, a more thorough analysis of the flowering behaviors of the various mutants and transgenic lines has been done. However, the new flowering time data has some inconsistencies that interfere with the rather singular idea the authors have of SSF stability influencing FLC expression. In LDs, the days to bolting and the leaf numbers match quite well and with published data of *f1c-3* mutants. However, in SDs, *ssf* flowers much earlier in terms of days to bolting (20-25 days) whereas the leaf numbers are quite similar to Col-0 (4 leaves). In the literature, *f1c-3* mutants tend to flower 7-17 leaves and 8 days earlier in SDs compared to Col-0. Therefore, the leaf initiation rate in SDs appears to be much higher in *ssf*. The summary statement “both mutants showed early flowering and decreased FLC expression in both long-day and short-day growth conditions” (Line 135-136) ignores the differences between leaf numbers and days, which should be acknowledged.

Response: We are grateful for the Reviewer’s comments. We agree with the opinion that under short-day growth conditions, the leaf initiation rate might be higher in *ssf* mutants because compared with wild type Col-0, both *ssf-1* and *ssf-2* plants indeed showed a large difference in days to bolting and a relatively small difference in leaf number (Fig. 2b, c, Supplementary Fig. 3b-e). We have added this point the text. Please refer to the unmarked manuscript, lines 149-155.

Furthermore, the authors do not assay FLC levels in SDs, which is suggested in the summary sentence. This should also be changed to accurately reflect the results shown.

Response: Thank you. We agree and have assayed the *FLC* expression under short-day growth conditions. The results showed that *FLC* was downregulated in the *ssf* mutant, consistent with the data under long-day conditions. The results have been added to the revised manuscript. Please refer to the new Fig. 2e and the unmarked manuscript, lines 147-149.

These phenotypes in SDs suggest that there is much more going on in the *ssf* mutant than modulating FLC expression alone. Although the authors acknowledge this, I would suggest a more explicit statement in the abstract that highlights the putative complex nature of the molecular mechanism underlying the *ssf* phenotype and directs the reader to the identification of a specific example of how modifications of SSF activity have been selected during evolution to modify FLC levels in LDs and possibly SDs.

Response: We agree with these valuable comments. Due to the space limitations of the abstract, we have incorporated this statement in the Results section to show the putative complex nature of

the *ssf* phenotype. Please refer to the unmarked manuscript, lines 154-155.

A comment that has not been sufficiently addressed is the statistical analysis. A student's t-test can come in many forms. In the methods, an explicit statement of whether a one-tailed, two-tailed, paired or unpaired test was performed is essential to evaluate the appropriateness of the test.

Response: We apologize and have added the explicit statement, "Asterisks indicate significant differences, two-tailed unpaired *t*-test" in the legends of all the figures. Please also find the description of the *t*-test in the Methods section, lines 446-448.

Reviewer #2 (Remarks to the Author):

My concerns have been adequately addressed in the revision.

Response: Thank you very much for your valuable comments.

Reviewer #3 (Remarks to the Author):

I read with interest the revised version of the manuscript from Peijin Li's group. Just to re-iterate, I had in my earlier review indicated that the manuscript clearly shows 1) Mutations in *SSF* genes appear to cause early flowering. 2) *SSF* interacts with *CUL1* and 3) The polymorphism in *SSF1* (N414D) potentially impacts the degradation of *SSF1*. The authors now also show the *SSF1* polymorphism (N414D) in natural condition is associated with a difference in flowering time (Supp Fig 9a).

I had raised two major concerns earlier. First, I argued that there is no evidence in the GWAS analysis to focus on *SSF1*. The authors now respond saying that if they increase the minor allele frequency to 15% (i.e., minor allele count of 15 out of 102 lines) in rebuttal it says 15%, in the paper it says 18% the all other noise disappears but the peak around *SSF1* remains (Fig 1a). I can't see it in their Figure whether this is significant at all. I do not have a problem in their *SSF1* analysis. But arguing that it came through GWAS is in my view simply spinning the story and not really supported by evidence. I am also not sure why they have to justify using this approach. Rather, simply state that we looked for regions in the GWAS that are potentially harbouring candidate genes and state while there was not a significant peak, there was a potential association (I guess it will turn out to be significant in a simple linear reg, without correcting for pop structure) around *SSF1*, one of the *FCA* orthologues. This would be a more accurate description than their story spinning, which I do not support.

Response: We appreciate the Reviewer's comment. We agree and have rewritten this section being more circumspect about the GWAS evidence that led us to pursue *SSF* as a candidate gene. We have also double-checked the MAC (which is indeed 15 in the analyses presented herein, so we apologize for the confusion). We also agree and have described the GWAS with population structure correction more cautiously by stating, "While no SNPs were significantly associated with the phenotype after Bonferroni multiple testing correction (Fig. 1a), we nonetheless looked for potential candidate genes under GWAS peaks that stood out against the background." In addition, according to the reviewer's comments, we also performed GWAS analysis without

population structure correction, which showed a significant association at the At2g47310 region close to the end of chromosome 2, as expected (Supplementary Fig. 1a).

The initial decision to pursue *SSF* as a candidate gene in fact arose from a combination of the GWAS and QTL results, and the *SSF-N414D* identified between the two parental accessions (Ull2-5 and Lov-1) of the F2 provided an important basis for the subsequent functional analysis of *SSF*. Consequently, we think it better to include the GWAS and QTL results as a starting point of the story. Please refer to the GWAS with population correction in Fig. 1a, GWAS with linear regression (no population correction) in Supplementary figure 1a, and the unmarked manuscript, lines 98-109. The methods have also been updated (lines 439-443).

My second concern was that they are “reproducing” a figure from their previous publication without properly citing it. Even now they “reproducing” a figure calling it “reanalysing”... There is no reanalysis. Fig 1b and Supplementary Fig 1b and direct “reproduction” of the Figure published by them in *Genes and Development* (Li et al, *Genes and Dev*, 2014, Fig 2D, panel titled ULL2-5 x Lov-1 SV2 x SV1). I also do not understand, why they simply do not state this explicitly that it is a reproduction of a previously published data. The authors simply stat that this QTL has not been studied in depth previously and cite their previous work, while describing all the methods as if it is a new analysis, which is simply not true. I find this very disturbing.

Response: We are grateful for the Reviewer’s comments, which have been very important for improving the accuracy of our manuscript. We agree and have removed the old Fig. 1b, Supplementary figure 1b, and the QTL analysis methods, and re-written the text by citing the *Genes and Dev* 2014 paper to show the context of the story. Please refer to the new Fig. 1, Supplementary figure 1, and the unmarked manuscript, lines 115-120.

I also disagree with their assertion that if you group your plants on the basis of Chr2 that the chr 5 QTL alleles will be randomly distributed. This is not correct. Typically, ghost QTLs appear when you do not correct for the effect of one QTL. A proper way of doing the experiment would be to genotype the same plants for the Chr 5 QTL and correct for that variation and then ask whether the Chr 2 allelic difference remains. (which it might).. But shrugging it off saying that they will be randomly distributed is not accurate. The key question is whether the expression difference observed is primarily driven by Chr 2 allele or by the LD or any non-random distribution of the Chr 5 alleles given they explain most of the variation, while Chr 2 allele in this cross primarily accounts for only 10% of the variation.

Response: Thank you for this valuable comment. We agree and performed the plant genotyping and screening work as recommended, in an attempt to correct the phenotypic variation mediated by Chr5 QTL and isolate the effect of Chr2-QTL. We genotyped the two QTLs on chromosome 5 (Chr5-QTL1: at the top of chromosome 5; Chr5-QTL2: at the bottom of chromosome 5) with markers mostly close to the two QTL peaks, respectively, to screen for the F2 lines with common Chr5-QTLs. Then, these plants were again genotyped with the At2g47310 marker for Chr2-QTL to search for two groups of lines (Chr2-QTL/Ull2-5 and Chr2-QTL/Lov-1) for RNA extraction and *FLC* expression analysis.

By using this method, we genotyped the two Chr5-QTLs of 279 F2 lines and found nine combinations (please see the table and note below). To more reliably reveal the contribution of Chr2-QTL, we chose the combination that had the largest number of lines (both Chr5-QTL1 and

Chr5-QTL2 are heterozygous, which have 65 lines) for further analysis. These 65 lines were then divided into three subgroups (17 lines with Chr2-QTL/Ull2-5, 19 lines with Chr2-QTL/Lov-1, and 29 Chr2-QTL/Hets) with the At2g47310 marker for Chr2-QTL. We used Chr2-QTL/Ull2-5 and Chr2-QTL/Lov-1 lines for gene expression analysis. The RNA of these lines was extracted separately, and qRT-PCR analysis indicated that in the common Chr5-QTL background, the Chr2-QTL/Lov-1 lines presented higher *FLC* expression than the Chr2-QTL/Ull2-5 lines (Fig. 1c). These results were consistent with the increased flowering time attributed to the Lov-1 allele of Chr2-QTL (Fig. 1d). Since all the F2 lines included in this analysis had a common genotype at both chromosome 5 QTLs, the observed expression differences were likely primarily attributable to the QTL on chromosome 2. Please refer to the new Fig. 1c, d and unmarked manuscript, lines 123-130.

The methods employed for F2 genotyping, background correction, and *FLC* expression analysis have been included in the Methods section. Please refer to the unmarked manuscript, lines 459-468.

Table: Genotyping and line number of the QTL in the F2 population (Ull2-5×Lov-1):

Genotype of QTLs	Line number	
BBCCAA	4	
BBCCaa	5	
BBCcAA	10	
BBCcaa	9	
BBccAA	2	
BBccaa	3	
BbCCAA	8	
BbCCaa	9	
BbCcAA	17	Selected for FLC comparison
BbCcaa	19	
BbccAA	6	
Bbccaa	5	
bbCCAA	6	
bbCCaa	6	
bbCcAA	10	
bbCcaa	11	
bbccAA	6	
bbccaa	7	
BBCCAa	7	Chr2-QTL is heterozygous
BBCcAa	13	
BBccAa	8	
BbCCAa	23	
BbCcAa	29	
BbccAa	8	
bbCCAa	14	
bbCcAa	22	

bbccAa	12	
Note:	Chr2-QTL: A/a	Abbreviation of the three QTLs
	Chr5-QTL1: B/b	
	Chr5-QTL2: C/c	

Reviewer #1 (Remarks to the Author):

The authors have adequately addressed my comments. Congratulations to the authors on a nice paper.

[Editor: Reviewer #3 is unavailable. We asked Reviewer #1 to comment on your responses to Reviewer #3's previous suggestions. Reviewer #1 states in Remark to Editor section that Reviewer #3's previous concerns have been adequately addressed.]

Dear Dr An and Reviewers:

Thank you very much for your email and the comments our manuscript titled "Molecular variation in a functionally divergent homolog of FCA regulates flowering time in Arabidopsis thaliana" (ID: NCOMMS-20-06055B)." All the comments were valuable and very helpful for revising and improving the quality of our paper. We have studied the comments carefully and addressed each one, and we hope that our manuscript now meets with the approval.

Figures 5, 6, 7 and Supplementary Fig. 3, 4, 5, 7 and all the legends and Supplementary data have been updated, and the revised portion of the text has been marked with the tracking tool in Word. The main corrections in the paper and the responses to the reviewer's comments are supplied in the "point-by-point response" file and "Attached document" file.

We would like take this chance to show our appreciation again to you, other editors and all the reviewers, and we look forward to hearing from you soon.

With best regards,

Peijin

REVIEWERS' COMMENTS

Reviewer #1 (Remarks to the Author):

The authors have adequately addressed my comments. Congratulations to the authors on a nice paper.

[Editor: Reviewer #3 is unavailable. We asked Reviewer #1 to comment on your responses to Reviewer #3's previous suggestions. Reviewer #1 states in Remark to Editor section that Reviewer #3's previous concerns have been adequately addressed.]

Response: Thank you very much for your valuable comments. All of the comments have been very helpful for improving our manuscripts.